# Diffusion-Driven Two-Stage Active Learning for Low-Budget Semantic Segmentation

**Jeongin Kim**[1]  **Wonho Bae**[2]  **YouLee Han**[1]  **Giyeong Oh**[3]
**Youngjae Yu**[4]  **Danica J. Sutherland**[2,5]  **Junhyug Noh**[1]*

[1]Ewha Womans University   [2]University of British Columbia
[3]Yonsei University   [4]Seoul National University   [5]Amii

{jn.kim, oneglass, junhyug}@ewha.ac.kr   {whbae, dsuth}@cs.ubc.ca
hard2251@yonsei.ac.kr   youngjaeyu@snu.ac.kr

## Abstract

Semantic segmentation demands dense pixel-level annotations, which can be prohibitively expensive – especially under extremely constrained labeling budgets. In this paper, we address the problem of low-budget active learning for semantic segmentation by proposing a novel two-stage selection pipeline. Our approach leverages a pre-trained diffusion model to extract rich multi-scale features that capture both global structure and fine details. In the first stage, we perform a hierarchical, representation-based candidate selection by first choosing a small subset of representative pixels per image using MaxHerding, and then refining these into a diverse global pool. In the second stage, we compute an entropy-augmented disagreement score (eDALD) over noisy multi-scale diffusion features to capture both epistemic uncertainty and prediction confidence, selecting the most informative pixels for annotation. This decoupling of diversity and uncertainty lets us achieve high segmentation accuracy with only a tiny fraction of labeled pixels. Extensive experiments on four benchmarks (CamVid, ADE-Bed, Cityscapes, and Pascal-Context) demonstrate that our method significantly outperforms existing baselines under extreme pixel-budget regimes. Our code is available at
`https://github.com/jn-kim/two-stage-edald`.

## 1   Introduction

Semantic segmentation is a core task in computer vision: assign a class label to each pixel in an image. Applications abound in areas such as autonomous driving, robotics, and medical image analysis. Despite the impressive performance of convolutional [1, 2] and Transformer-based models [3–5], an overarching challenge remains: *how do we obtain the pixel-level annotations required to train these models effectively?* Large-scale datasets like Cityscapes [6] and Pascal-Context [7] require meticulous per-pixel labeling, rendering data collection *extremely* costly.

Active learning (AL) aims to mitigate labeling costs by strategically selecting a subset of unlabeled data points, for example, pixels, for annotation [8–10]. However, traditional uncertainty-based AL methods for segmentation often select redundant pixels, since the $k$ most uncertain pixels are often near one another [11, 12]. Representation-based methods do not select redundant pixels, but they tend to miss informative pixels near the boundary of objects. Furthermore, as representation-based methods usually require computation of the pairwise similarity of candidates, it is prohibitive to consider all the available pixels at once.

---

*Corresponding author

In real-world scenarios, the total number of pixels is massive; for example, $N$ images each of resolution $H \times W$ yield a total of $N \cdot H \cdot W$ pixels. However, because of the high annotation cost, the labeling budget can be orders of magnitude smaller. We therefore formalize a *low-budget active learning* setting for semantic segmentation: at each AL round, one may annotate only $b$ pixels from the entire unlabeled pool, with $b \ll N \cdot H \cdot W$. In our standard (extreme) regime, we allocate *one pixel per image* for 10 rounds – *i.e.*, $b = 0.1N$ pixels per round and a total of $\approx N$ pixel labels across the whole process.

A promising direction to address this data-scarce labeling scenario is to harness the representation power of *diffusion models* [13, 14]. Diffusion models are able to iteratively denoise over multiple time steps, generating highly detailed images [13]. The initial steps of this reverse diffusion process capture the overall structure of objects, while the later steps focus on generating fine-grained details. Baranchuk et al. [15] demonstrate that multi-timestamp features from diffusion models are beneficial for semi-supervised segmentation tasks. We will show that they can be used for uncertainty estimation, due to their ensemble-like nature.

We propose a two-stage strategy that efficiently pinpoints both diverse and epistemically uncertain pixels. In **Stage 1**, we employ a representation-based AL method, in particular MaxHerding [16], to extract a set of candidate pixels which is manageable in size yet representative; we first narrow down candidates in each image, then refine across all images to ensure global diversity. In **Stage 2**, we exploit the diffusion backbone's stochastic multi-scale features to compute an *entropy-augmented disagreement* score (eDALD): we measure mutual information between noisy feature samples and labels, then add a single-sample entropy term to capture prediction confidence. This combined criterion prioritizes pixels that both lie in under-explored regions of feature space and carry high model uncertainty, maximizing the benefit of each annotation under an extremely limited pixel budget.

Our contributions are summarized as follows:

- We formalize and address the challenging *low-budget active learning for semantic segmentation*, where a mere fraction of pixels can be annotated per round.
- We introduce a scalable two-stage pipeline (coverage $\rightarrow$ uncertainty) for pixel-level AL: a *local-then-global* MaxHerding stage yields a representative candidate pool, followed by uncertainty-based refinement.
- We develop a diffusion-native uncertainty criterion (eDALD) that combines disagreement from stochastic *multi-timestep* features with a single-pass entropy term; *used after coverage*, it complements MaxHerding and yields substantial gains over one-stage uncertainty-only or coverage-only variants under tiny budgets.
- We present *comprehensive experiments* on benchmark datasets – CamVid, ADE-Bed, Cityscapes, and Pascal-Context – demonstrating consistent gains over multiple baselines when budgets are severely constrained.

## 2 Related Work

### 2.1 Active Learning for Classification

Active learning (AL) is a learning framework to improve the data efficiency of training machine learning models by strategically selecting the most informative data points for annotation. AL methods can be broadly categorized into two main approaches: uncertainty-based methods and representation-based methods.

**Uncertainty-based Methods.** Uncertainty-based methods [17–20] generally select data points near the decision boundary. These simple methods are generally effective; however, they can be prone to selecting pixels with erratic or extreme predicted class probabilities. These pixels, such as isolated noise artifacts or very rare patterns, often exhibit high uncertainty yet lie far from the true decision boundary, contributing little to model improvement. To avoid this problem, some work has attempted to measure the change of a model for a given candidate data point and its corresponding pseudo-label [21–23] or the change of model outputs [24–27]; in practice, these methods typically do not perform much better than the simple ones when using predictors defined by deep networks.

From a Bayesian perspective, on the other hand, the goal of AL is essentially to select data points that maximize mutual information (or equivalently, information gain) between model parameters $\theta$

and the true label $Y$ to be observed, given an input $x$ and a model $f$.

$$\tilde{x}^* \in \underset{x \in \mathcal{U}}{\arg\max} \, I(\theta; Y \mid x) \quad \text{where} \quad I(\theta; Y \mid x) = H(Y \mid x) - \underset{\theta}{\mathbb{E}}\Big[H(Y \mid \theta, x)\Big] \tag{1}$$

This objective is known as Bayesian Active Learning by Disagreement (BALD), since it selects a data point where model parameters under the posterior distribution disagree the most for the output label [28]. PowerBALD [29] avoids redundant top-$k$ selection by sampling without replacement from a "powered" BALD distribution $p(i) \propto s_i^{\beta}$. Balanced Entropy (BalEnt) and its acquisition variant BalEntAcq [30] fit each softmax marginal to a Beta distribution, producing a bounded score that balances epistemic and aleatoric uncertainty.

**Representation-based Methods.** These methods instead select samples that are representative of the entire unlabeled data pool, often by analyzing learned data embeddings or features [11, 31]. Core-set methods, for example, aim to find a small subset of the unlabeled data that best covers the diversity of the entire dataset in a feature space [9, 31, 32]. Another notable approach is Variational Adversarial Active Learning (VAAL), which employs a variational autoencoder and a discriminator to identify samples that are most distinct from the already labeled data in a learned latent space [32, 33].

Hacohen et al. [34] recently demonstrated that representation-based methods perform significantly better than uncertainty-based methods in low-budget regimes. A sequence of works has proposed improving notions of coverage to select better representing samples: probability coverage [35], generalized coverage [16] and uncertainty coverage [36].

## 2.2 Active Learning for Semantic Segmentation

Active learning for semantic segmentation extends classification – style AL to the pixel-wise labeling domain, where annotation costs are dramatically higher. Unlike classification, every pixel's label matters, so AL methods must balance selection granularity, spatial coherence, and computational efficiency. Broadly, existing approaches fall into three categories – image-level, region-level, and pixel-level – each trading off annotation cost against precision and implementation complexity.

**Image-level Methods.** The most straightforward adaptation of classification-style active learning to segmentation is to select entire images for annotation. Many image-level selection strategies for semantic segmentation directly adapt techniques from classification. For instance, uncertainty-based methods often calculate a per-pixel uncertainty score and then aggregate these scores across the entire image, such as by averaging, to decide which images to query [11]. Xie et al. [31] also employ image-level selection, though they enhance it by considering semantic difficulty. Similarly, representation-based approaches like Core-set or VAAL can be applied by selecting images whose overall feature representations contribute most to dataset diversity or are most distinct from already labeled data [31, 33]. While simple to implement, image-level querying quickly exhausts annotation budgets because every pixel – even in well-understood regions – must be annotated.

**Region-level Methods.** To reduce per-query cost, region-level approaches annotate clusters of pixels – such as superpixels [37, 38], rectangular patches [39], or bounding boxes [40]. These methods save clicks by labeling chunks at once and often combine uncertainty with coverage or learned difficulty. However, they assume a minimally competent model to estimate region scores: under extreme low budgets ("cold-start"), unreliable uncertainty or difficulty estimates can misrank regions and degrade performance. Moreover, multi-class regions can introduce label ambiguity when boundaries cross cluster edges.

**Pixel-level Methods.** Pixel-level querying – selecting individual pixels – incurs the lowest per-unit cost and avoids region-boundary ambiguity but has received less attention, as each query yields limited information. PixelPick [41] applies margin sampling to individual pixels, and Didari et al. [42] use BalEntAcq [30] as a Bayesian pixel-uncertainty measure. Purely uncertainty-driven pixel selection, however, demands many queries and often fails in low-budget regimes.

In contrast to these prior works, we introduce the first practical framework for *low-budget* active learning in semantic segmentation. Our two-stage pipeline first uses representation-based sampling to build a compact, diverse candidate pool, then applies uncertainty-driven selection to pick the final pixels – achieving high segmentation accuracy with only a handful of annotated points in low-budget scenarios.

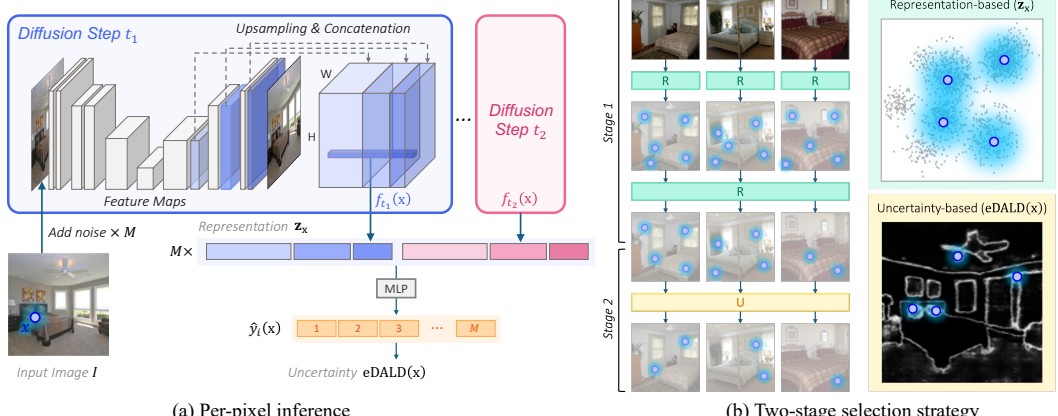

(a) Per-pixel inference        (b) Two-stage selection strategy

Figure 1: Overview of the proposed pipeline (illustrative hyperparameters). (a) For each pixel $x$, the pre-trained diffusion model yields a feature representation $\mathbf{z}_x$, an uncertainty estimate $\mathrm{eDALD}(x)$, and the class prediction $\hat{y}(x)$ (here shown with $T = 2$ diffusion steps and $L = 3$ layers purely for illustration). (b) These outputs drive our two-stage selection: Stage 1 picks $K = 5$ representative pixels per image to form a global pool of size $M = 10$, then Stage 2 applies the uncertainty criterion to select the final $b = 5$ pixels (all parameter values are example settings, not the exact experimental hyperparameters).

## 2.3 Diffusion Models in Vision

**Generative Diffusion.** Denoising Diffusion Probabilistic Models (DDPM) [13] and score SDE [43–45] introduce a novel paradigm of generative models: progressively destroy an image by adding noise, then learn to gradually reverse this process. Follow-up works such as Latent Diffusion Models (LDM) [14] improve scalability for high-resolution images.

**Diffusion for Downstream Tasks.** Some recent research explore leveraging the representations from diffusion models for tasks beyond generation [46]. In particular, medical image segmentation [47] and few-shot segmentation [15] benefit from multi-scale features spanning from broad global context (earlier denoising steps) to detailed object boundaries (later steps). Nevertheless, the integration of diffusion representations into *active learning* remains unexplored.

## 3 Proposed Method

In this work, we present a novel two-stage active learning pipeline tailored for semantic segmentation under a strictly low-budget setting. Our approach integrates the strengths of pre-trained diffusion models for feature extraction with a two-pronged sample selection strategy that first ensures representation diversity and then refines the selection using uncertainty estimation. The overall pipeline is illustrated in Figure 1 and detailed in Algorithm 1.

### 3.1 Preliminaries

**Problem Setup.** Let $\mathcal{I} = \{I^{(n)}\}_{n=1}^{N}$ denote a set of $N$ images, where each image $I_i$ contains $H \times W$ pixels. At each AL round $r \in [R]$ with a labeled pixel set $\mathcal{L}_r$ and an unlabeled set $\mathcal{U}_r$, we select a subset $\mathcal{S}_r \subseteq \mathcal{U}_r$ such that $|\mathcal{S}_r| = b$ for annotation, where $b$ is a pre-defined labeling budget per round. More specifically, active learning involves the following steps in each $r$-th round:

1. Measure how informative each pixel in (or a subset of) $\mathcal{U}_r$ is
2. Select a subset $\mathcal{S}_r \subseteq \mathcal{U}_r$ that consists of $b$ "most informative" pixels
3. Update both $\mathcal{L}_{r+1} \leftarrow \mathcal{L}_r \cup \mathcal{S}_r$ and $\mathcal{U}_{r+1} \leftarrow \mathcal{U}_r \setminus \mathcal{S}_r$
4. Retrain the segmentation model on $\mathcal{L}_{r+1}$

In this work, we focus on extremely low-budget setting *i.e.*, $b \ll (N \times H \times W)$, the total number of pixels. The challenge is then to select highly informative pixels for robust generalization in semantic segmentation tasks.

---
**Algorithm 1** Two-Stage Low-Budget Active Learning at Round $r$
---
**Require:** Image set $\mathcal{I}$, labeled pixel set $\mathcal{L}_r$, unlabeled pixel set $\mathcal{U}_r$, diffusion feature extractor $f$, current segmentation head $s_\theta$, the number of candidates per image $K$, global candidate pool size $M$, per-round budget $b$.
**Ensure:** Update labeled set $\mathcal{L}_r$ and unlabeled set $\mathcal{U}_r$ with $b$ new annotated pixels.

 1: $\mathcal{M}_0 \leftarrow \emptyset$  // Initialize the global candidate pool
 2: // Stage 1: Representation-based Candidate Selection
 3: **for** each image $I \in \mathcal{I}$ **do**
 4:     Extract multi-scale features $\mathcal{Z} = \{\mathbf{z}_x\}_{x \in I, x \in \mathcal{U}}$ using $f$
 5:     Select a set of $K$ representative pixels $\mathcal{R}$ by applying MaxHerding to $\mathcal{Z}$
 6:     $\mathcal{M}_0 \leftarrow \mathcal{M}_0 \cup \mathcal{R}$
 7: **end for**
 8: Obtain the global candidate pool $\mathcal{M}$ of size $M$ by applying MaxHerding to $\mathcal{M}_0$
 9: // Stage 2: Uncertainty-driven Selection
10: **for** each $x \in \mathcal{M}$ **do**
11:     Compute eDALD(x) as in Eq. (8) using the segmentation head $s_\theta$ and $f$
12: **end for**
13: $\mathcal{B}_r \leftarrow$ Top-$b$ pixels from $\mathcal{M}$ sorted by descending eDALD(x)
14: $\mathcal{L}_{r+1} \leftarrow \mathcal{L}_r \cup \mathcal{B}_r, \mathcal{U}_{r+1} \leftarrow \mathcal{U}_r \setminus \mathcal{B}_r$
15: **return** $\mathcal{L}_{r+1}, \mathcal{U}_{r+1}$
---

**Diffusion-based Semantic Segmentation.**   Our segmentation framework builds on the LEDM architecture [15] and exploits a pre-trained diffusion model denoted as $f(\cdot)$ – *e.g.*, DDPM [13] to extract robust multi-scale features. Note that it is crucial to utilize good representations, particularly in low-budget regimes, as a segmentation model trained on a small labeled set itself would not be sufficient to learn good representations.

For an image $I$, the diffusion model produces feature maps at multiple denoising timesteps $t \in \{t_1, t_2, \ldots, t_T\}$ and layers $l \in \{l_1, l_2, \ldots, l_L\}$. For a given pixel x in an image $I$ (at coordinates $(w, h)$, where $1 \leq w \leq W$ and $1 \leq h \leq H$), let $f_{t,l}(x) \in \mathbb{R}^{D_{t,l}}$ denote a feature vector from a layer $l$ at denoising timestep $t$. We obtain a *multi-scale* representation by concatenating these features:

$$\mathbf{z}_x := \Big[ f_{t_1,l_1}(x); \ldots; f_{t_1,l_L}(x); \ldots; f_{t_T,l_1}(x); \ldots; f_{t_T,l_L}(x) \Big] \in \mathbb{R}^D, \qquad (2)$$

where $D = \sum_{i=1}^{T} \sum_{j=1}^{L} D_{t_i,l_j}$. This comprehensive feature vector $\mathbf{z}_x$ encapsulates rich semantic cues learned during the diffusion process as demonstrated in [15]. We omit the subscript x when it is clear from context.

The predicted probability of segmentation for $C$ object classes is produced by a lightweight segmentation head $s_\theta : \mathbb{R}^D \to \mathbb{R}^C$. It consists of a 2-layer MLP with a ReLU activation and batch normalization, followed by a softmax layer. Note that we deliberately add a parameter notation $\theta$ only for the segmentation head $s_\theta$ not for the diffusion model $f$, to describe that we only update the segmentation head not the diffusion model.

### 3.2   Representation-based Candidate Selection

Representation-based AL methods rely on the pairwise similarity measure. Hence, it is often infeasible to select pixels from all possible $(N \times H \times W)$ pixels, as it requires $O(N^2 W^2 H^2)$ computation for a pairwise comparison. To tackle this problem, we adopt a *two-step* strategy that leverages the MaxHerding algorithm [16], a representation-based AL method, in each step to ensure diversity while reducing computational complexity; please refer to Line 1–8 in Algorithm 1.

We first identify $K$ representative pixels *within each image $I$*. More specifically, for each pixel $x \in I$, we obtain a multi-scale feature vector $\mathbf{z}$ extracted from the diffusion model $f$ using Eq. (2). We then greedily select an optimal pixel $\tilde{x}^*$ at a time using generalized coverage $\hat{C}_k$ [16] as follows:

$$\tilde{x}^* \in \underset{\tilde{x} \in I, \tilde{x} \in \mathcal{U}}{\operatorname{argmax}} \hat{C}_k(\mathcal{L} \cup \{\tilde{x}\}) \ \text{ where } \ \hat{C}_k(\mathcal{L} \cup \{\tilde{x}\}) := \frac{1}{|\mathcal{U}|} \sum_{x \in \mathcal{U}} \Big[ \max_{x' \in \mathcal{L} \cup \{\tilde{x}\}} k(x, x') \Big]. \qquad (3)$$

Here, $k$ is a function that measures pairwise similarity between x and x'. We simply use a RBF kernel for $k$ as $k(\mathrm{x}, \mathrm{x}') = \exp\left(-\frac{\|\mathbf{z}_\mathrm{x} - \mathbf{z}_{\mathrm{x}'}\|_2^2}{\sigma^2}\right)$. We repeat this until we obtain $K$ pixels yielding a compact yet diverse subset $\mathcal{R}$ for each image $I$.

By restricting the search space for selecting pixels to an individual image, we reduce the size of candidates to $(H \times W)$ compared to $(N \times H \times W)$. We obtain the initial global pool $\mathcal{M}_0$ by merging the selected subset $\mathcal{R}$ for each image $I$. This pool has size $(N \times K)$, which is much smaller than $(N \times H \times W)$. To further refine these candidates, we apply MaxHerding again *across the entire merged set* to select $M$ representative pixels, which form a candidate pool for the final $b$ pixels. The final global pool $\mathcal{M}$ of size $M$ thus achieves good coverage over all images collectively. Typically, $M \ll N \cdot H \cdot W$.

Through this stage, we ensure that our final candidate set $\mathcal{M}$ is both *diverse* (capturing the data manifold effectively) and *manageable in size*, laying the foundation for more targeted selection via uncertainty in the next step.

### 3.3 Uncertainty-driven Selection

**Diffusion-based Active Learning by Disagreement (DALD).**  From the diverse candidate pool $\mathcal{M}$, we further refine our selection by identifying the most informative $b$ pixels using an uncertainty-based measure; refer to Line 9–13 in Algorithm 1. Our method uniquely exploits the structure of the diffusion model: each pixel x is associated with multiple feature vectors.

Inspired by the BALD objective in Eq. (1), we propose *Diffusion-based Active Learning by Disagreement* (DALD) that selects a new data point as follows:

$$\mathrm{x}^* \in \underset{\mathrm{x} \in \mathcal{U}}{\arg\max} \; \mathrm{I}(\hat{Y}; Z \mid \mathrm{x}, s_\theta, f) \tag{4}$$

where $\hat{Y}$ denotes a random variable for the predicted label of a pixel x and $Z$ denotes a random variable for the concatenated multi-scale features from x computed using Eq. (2). Note that the stochasticity of $Z$ comes from noise added to x at $t_1, \ldots, t_T$ timesteps. We define this conditional distribution of $Z$ given x as $q(\cdot \mid \mathrm{x}; f)$. Furthermore, suppose $X$ denotes a random variable for a pixel x, then the computational probabilistic graphical model is defined as $X \to Z \to \hat{Y}$, assuming that $X$ does not provide additional information about $\hat{Y}$ knowing $Z$.

The motivation of DALD is to select a data point where information gain for a noised multi-scale feature $Z$ is maximized given a label $y$. Intuitively, if the information gain of a multi-scale feature is close to 0, knowing the label $y$ does not help removing the ambiguity of the feature,[1] while if the information gain of a multi-scale is high, knowing the label $y$ significantly removes the uncertainty of the noised feature.

The mutual information in Eq. (4) can be decomposed into the following two terms:

$$\mathrm{I}(\hat{Y}; Z \mid \mathrm{x}, s_\theta, f) = \underbrace{\mathrm{H}(\hat{Y} \mid \mathrm{x}, s_\theta, f)}_{\text{Unconditional entropy}} - \underset{\mathbf{z} \sim q(\cdot | \mathrm{x})}{\mathbb{E}}\left[\underbrace{\mathrm{H}(\hat{Y} \mid Z = \mathbf{z}, \mathrm{x}, s_\theta)}_{\text{Conditional entropy}}\right]. \tag{5}$$

We compute the uncertainty of the predicted label $\hat{Y}$ instead of a multi-scale feature $Z$ to make computation easy.

The computation of conditional entropy is relatively more straightforward. As $\hat{Y}$ is conditionally independent to a clean pixel x given a multi-scale feature $\mathbf{z}$, the expected conditional entropy can be approximated using $M$ samples of $\mathbf{z}$, as follows:

$$\underset{\mathbf{z} \sim q(\cdot | \mathrm{x})}{\mathbb{E}}\left[\mathrm{H}(\hat{Y} \mid Z = \mathbf{z}, \mathrm{x})\right] \approx -\frac{1}{M} \sum_{m=1}^{M} \sum_{y \in \mathcal{Y}} \hat{p}_\theta(y | \mathbf{z}^{(m)}) \cdot \log \hat{p}_\theta(y | \mathbf{z}^{(m)}) \tag{6}$$

---

[1] It can happen either when aleatoric uncertainty is very high or epistemic uncertainty is very low.

where $\hat{p}_\theta(y|\mathbf{z}) = s_\theta(\mathbf{z})_y$. Therefore, conditional entropy is approximated as the mean of entropy. In contrast, we compute the unconditional entropy as the entropy of the mean predictions as follows:

$$\mathrm{H}(\hat{Y} \mid \mathrm{x}, s_\theta, f) \approx -\sum_{y \in \mathcal{Y}} \bar{p}_\theta(y|\mathrm{x}) \cdot \log \bar{p}_\theta(y|\mathrm{x}), \ \text{where} \ \ \bar{p}_\theta(y|\mathrm{x}) = \frac{1}{M} \sum_{m=1}^{M} \hat{p}_\theta(y|\mathbf{z}^{(m)}). \quad (7)$$

**Entropy-Augmented DALD (eDALD).**   Although disagreement-based selection has the advantage of being less sensitive to aleatoric (or irreducible) uncertainty, it also has a clear limitation: it does not account for the confidence of model predictions. For example, two different segmentation models may output identical mutual information for a given pixel, although one may produce a low entropy (high confidence) output, while the other yields a high entropy (low confidence) output. This discrepancy illustrates that disagreement alone may fail to capture the full picture of predictive uncertainty.

As a simple and computationally inexpensive remedy to this limitation, we introduce an additional entropy term based on a separate sample. Specifically, our selection objective becomes the following:

$$\mathrm{x}^* = \underset{\mathrm{x} \in \mathcal{U}}{\mathrm{argmax}} \ \mathrm{eDALD}(\mathrm{x}), \ \text{where} \ \ \mathrm{eDALD}(\mathrm{x}) = \mathrm{I}\big(\hat{Y}; Z \mid \mathrm{x}, s_\theta, f\big) + \mathrm{H}\big(\hat{Y} \mid \mathbf{z}^{(0)}, \mathrm{x}\big). \quad (8)$$

where $\mathbf{z}^{(0)}$ denotes an independently drawn sample, separate from sample $m = 1, 2, \dots, M$ used to estimate the mutual information. This extra entropy term highlights pixels where the model is less confident, further enhancing the sensitivity of acquisition to both disagreement and absolute uncertainty.

### 3.4   Overall Selection and Training Procedure

Our complete active learning pipeline is summarized in Algorithm 1. At each of the $R$ rounds, we (1) extract multi-scale diffusion features $\mathbf{z}_\mathrm{x}$ for all unlabeled pixels $\mathrm{x} \in \mathcal{U}$, (2) select $b$ pixels through our two-stage sampler (MaxHerding $\rightarrow$ eDALD), (3) annotate their labels and update the labeled/unlabeled sets, and (4) train on the expanded labeled set.

During training, the segmentation head $s_\theta$ is optimized via a cross-entropy loss computed over the up-to-date labeled set $\mathcal{L}$:

$$\theta^* \in \underset{\theta}{\mathrm{argmin}} -\frac{1}{|\mathcal{L}|} \sum_{(\mathrm{x},y) \in \mathcal{L}} \log \hat{p}_\theta(y \mid \mathrm{x}, f).$$

This cycle is repeated over $R$ rounds, allowing the model to progressively learn from a carefully curated and highly informative set of pixels while keeping the annotation costs minimal.

## 4   Experiments

### 4.1   Datasets and Setup

We evaluate our low-budget active learning pipeline on four standard semantic segmentation benchmarks. All images are processed at $256 \times 256$ resolution for compatibility with the diffusion backbone:

- **CamVid** [48]: An urban driving dataset with 367 train and 233 test images, each labeled into 11 classes. All images are center-cropped to $256 \times 256$.
- **ADE-Bed:** A subset of ADE20K [49] consisting of bedroom images with 964 train and 650 test images annotated with the 30 most common object classes. We resize the shorter side to 256 pixels, preserve aspect ratio, then center-crop to $256 \times 256$.
- **Cityscapes** [6]: A street-scene dataset comprising 2,975 train and 500 validation images over 19 classes. We resize and center-crop to $256 \times 256$.
- **Pascal-Context** [7]: A scene parsing dataset providing dense semantic labels for more than 400 categories. Following convention [7, 50, 51], we use 33 most frequent categories. It contains 4,998 train and 5,105 validation images. Images are resized to $256 \times 256$ using bilinear interpolation.

**Budget Setting.**   We fix a total annotation budget $B$ to be the average of 1 labeled pixels per image, *i.e.*, $B = N$. For Pascal-Context with $N = 5{,}000$, this implies $B = 5{,}000$ pixels – a practical scale for large datasets. This budget is evenly split across $R = 10$ AL rounds, yielding $b = B/R = 0.1N$ pixels per round. As intended, we only annotate $0.0015\%$ of the total pixels after 10 AL rounds.

## 4.2 Implementation Details

**Diffusion Model.** We adopt an ImageNet pre-trained diffusion model [13] for feature extraction from a publicly available guided diffusion repo. To capture multi-scale cues, we sample $T = 3$ timesteps ($t_1 = 50, t_2 = 150, t_3 = 250$). At each timestep, we extract the features from $L = 4$ layers ($l_1 = 5, l_2 = 8, l_3 = 12, l_4 = 17$), resulting in a rich, concatenated representation for each pixel. For uncertainty estimation (DALD/eDALD), we draw $M = 5$ noisy feature samples per pixel.

**Candidate Selection.** Following Section 3.2, we apply MaxHerding [16] to the pixels of each image to obtain $K = 50$ representative samples, which are then merged across all images into an initial global pool $\mathcal{M}_0$. We further apply MaxHerding to $\mathcal{M}_0$ to obtain the final candidate set $\mathcal{M}$, which contains half of the samples in $\mathcal{M}_0$.

**Training Procedure.** We use Adam with an initial learning rate of $1 \times 10^{-3}$ for training the pixel classifier, with a batch size of 5. We apply early stopping if the segmentation loss does not improve for 50 consecutive iterations and the target pixel accuracy exceeds 95%. This training procedure is repeated for 10 active learning rounds, each time adding $b$ newly labeled pixels to the training set.

## 4.3 Quantitative Results

**Representation-First *vs*. Uncertainty-Only.** All results in Table 1 use our diffusion-backbone on CamVid dataset. We compare "UC Only'' (single-stage: uncertainty) against "Herding → UC" (two-stage: representation → uncertainty), reporting absolute and relative gains. Overall, enforcing diversity first consistently boosts simple criteria – entropy gains +5.51 mIoU (+21.81%), and margin sampling gains +1.5 mIoU (+4.8%) – confirming that representation filtering improves even basic uncertainty sampling.

However, pure DALD (random-noise) and BALD (MC-Dropout) actually worsen after MaxHerding (−2.76 and −1.8 mIoU, respectively), suggesting that disagreement-only measures can overemphasize noisy or redundant regions once diversity is already enforced. In contrast, their entropy-augmented versions show dramatic improvements: eBALD gains +6.16 mIoU (+23.73%), and eDALD gains +10.98 mIoU (+43.68%). This striking boost reflects the complementary roles of the two signals: disagreement identifies perturbation-sensitive pixels that entropy alone may overlook, while entropy recovers consistently low-confidence areas that disagreement misses. Their combination reshapes the acquisition ranking rather than merely scaling one criterion, leading to more balanced and informative selections. Eventually, "Herding → eDALD" achieves the highest mIoU of 36.12, demonstrating that combining disagreement with confidence-aware uncertainty, along with representation diversity, is crucial to achieve substantial gains under extreme low-budget settings.

Table 1: Effect of representation-first filtering on uncertainty sampling on CamVid, measured in mIoU (%). "UC Only" shows single-stage performance; "Herding → UC" adds MaxHerding before uncertainty. Gains are shown in absolute points and percentages, with positive values in blue and negative in red. All results are based on the mean ± std from three independent runs.

| Uncertainty | UC Only | Herding → UC | Gain (pp) | Gain (%) |
|---|---|---|---|---|
| Entropy | 25.26 ± 0.36 | 30.77 ± 0.44 | +5.51 | +21.81 |
| Margin | 31.27 ± 1.10 | 32.77 ± 0.75 | +1.50 | +4.80 |
| BALD | 24.59 ± 0.97 | 22.79 ± 0.89 | −1.80 | −7.32 |
| DALD | 23.81 ± 3.60 | 21.05 ± 1.05 | −2.76 | −11.59 |
| PowerBALD | 30.03 ± 0.76 | 31.57 ± 0.79 | +1.54 | +5.13 |
| PowerDALD | 31.30 ± 1.22 | 32.00 ± 0.66 | +0.70 | +2.24 |
| eBALD (Entropy + BALD) | 25.96 ± 1.92 | 32.12 ± 0.40 | +6.16 | +23.73 |
| eDALD (Entropy + DALD) | 25.14 ± 0.57 | 36.12 ± 0.24 | +10.98 | +43.68 |

**Performance Comparison with Baselines.** Table 2 compares mIoU after 10 rounds under a severe pixel-budget across four datasets. Pixel-level AL research is still nascent, so we benchmark against two representative methods – PixelPick [41] and Didari et al. [42] – both originally designed for settings where a fixed number of pixels per image can be labeled each round. In our extreme low-budget regime (on average only 0.1 pixels per image per round are labeled), we select the top-$b$

most uncertain pixels from the unified candidate pool across the dataset according to the acquisition criterion.

Given PixelPick is equivalent to Margin selection with DeepLabV3 backbone, the benefit of DDPM-backbone over DeepLabV3 can be estimated by comparing PixelPick (1st row) *vs.* Margin (5th row): DDPM-backbone improves the performance by $11.81$ mIoU on average ($22.85$ *vs.* $34.66$), and by up to $21.68$ mIoU on the ADE-Bed dataset ($8.35$ *vs.* $30.03$). Similarly, Didari et al. [42] (BalEntAcq + DeepLabV3, 2nd row) generally performs worse than BalEntAcq with DDPM-backbone (6th row). Putting all together, our proposed method, 2-Stage eDALD, achieves the best overall results, consistently outperforming all baselines on average by non-trivial margins (from $2.48$ to $17.36$ mIoU). This demonstrates that the entropy-augmented diffusion-based mutual information within a diverse candidate pool is notably effective under extreme low-budget regimes.

Table 2: mIoU (%) of active learning methods under a low-budget regime (10 rounds). Our two-stage method is highlighted in gray. All results are based on the mean $\pm$ std from three independent runs.

| Backbone | Method | CamVid | ADE-Bed | Cityscapes | Pascal-C | Avg |
|---|---|---|---|---|---|---|
| DeepLabV3 [52] | PixelPick [41] | $29.93 \pm 0.12$ | $8.35 \pm 0.41$ | $26.82 \pm 0.14$ | $26.28 \pm 0.09$ | 22.85 |
| | Didari et al. [42] | $22.47 \pm 0.10$ | $8.66 \pm 0.53$ | $19.85 \pm 0.07$ | $28.15 \pm 0.11$ | 19.78 |
| DDPM [13] | Random | $25.91 \pm 1.23$ | $17.83 \pm 0.62$ | $27.13 \pm 1.38$ | $41.70 \pm 2.08$ | 28.14 |
| | Entropy | $25.26 \pm 0.36$ | $23.02 \pm 1.64$ | $28.62 \pm 1.05$ | $42.09 \pm 1.99$ | 29.74 |
| | Margin | $31.27 \pm 1.10$ | $30.03 \pm 0.37$ | $32.23 \pm 1.21$ | $45.11 \pm 2.45$ | 34.66 |
| | BalEntAcq | $19.37 \pm 1.10$ | $17.48 \pm 1.36$ | $24.04 \pm 2.07$ | $33.06 \pm 4.18$ | 23.49 |
| | eDALD | $25.14 \pm 0.57$ | $23.06 \pm 1.29$ | $29.44 \pm 1.38$ | $43.05 \pm 0.12$ | 30.17 |
| | 2-Stage eDALD | $\mathbf{36.12 \pm 0.24}$ | $\mathbf{31.12 \pm 0.20}$ | $\mathbf{33.34 \pm 0.78}$ | $\mathbf{47.98 \pm 0.41}$ | **37.14** |

**Round-Wise Learning Curves.** Figure 2 plots mIoU over 10 AL rounds for various methods across four datasets. Similar to Table 2, PixelPick [41] and Didari et al. [42] use the DeepLabV3 backbone, whereas the others are based on DDPM. Across all benchmarks, our two-stage scheme (Herding $\rightarrow$ eDALD) consistently achieves the highest performance by the final rounds. In contrast, the single-stage uncertainty-based methods (Entropy, Margin, BalEntAcq, and eDALD) show only gradual improvements throughout the rounds, leading to comparatively smaller overall gains. Together, these results highlight the effectiveness of the proposed two-Stage eDALD strategy.

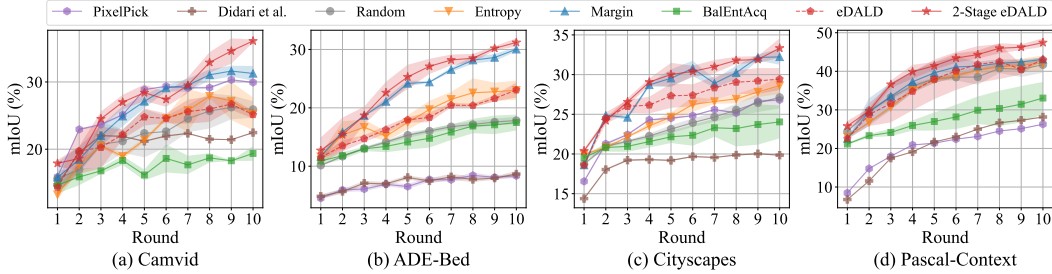

Figure 2: Active learning progression: mIoU (%) *vs.* AL rounds on (a) CamVid, (b) ADE-Bed, (c) Cityscapes, and (d) Pascal-Context. Curves compare Random sampling, single-stage uncertainty methods, and two-stage variants.

**Convergence to Fully-Supervised Performance.** We evaluate how quickly each method approaches fully supervised performance under a tiny per–round budget $b = 0.1N$. The fully supervised mIoUs are: ADE-Bed: $45.58$, CamVid: $52.22$, Cityscapes: $43.04$, Pascal-Context: $60.68$. Motivated by our low-budget findings and the adaptive-querying perspective, we use a simple two-phase schedule: rounds $1-10$ (the extreme low-budget phase) use our two-stage eDALD pipeline (coverage $\rightarrow$ uncertainty) to establish broad, non-redundant coverage; for rounds $> 10$ we drop the coverage stage and continue with uncertainty-only (Margin). This switch reflects two observations: (i) once coverage is established, diversity offers diminishing returns while residual errors concentrate near boundaries, where Margin is effective; and (ii) removing Stage 1 slightly reduces overhead while accelerating late-phase convergence.

Figure 3 shows mIoU *vs*. AL rounds at $b = 0.1N$. Across all four datasets, the proposed schedule consistently reaches $90\%$ of the fully supervised mIoU in far fewer rounds than PixelPick [41]. Concretely, two-stage eDALD (early) $\rightarrow$ Margin (late) attains the $90\%$ target within 21–47 rounds – specifically, CamVid: 32 rounds, ADE-Bed: 47, Cityscapes: 28, Pascal-Context: 21 – while labeling only $0.003\%-0.007\%$ of all pixels in total.

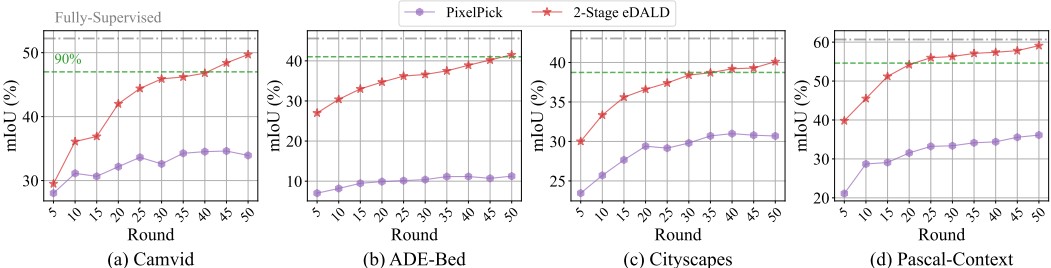

Figure 3: mIoU *vs*. AL rounds (budget $b = 0.1N$). Horizontal gray lines denote fully supervised mIoUs, and green lines denote the $90\%$ thresholds. We use two-stage eDALD for rounds $1-10$ and switch to Margin thereafter.

## 4.4 Qualitative Analysis

Figure 4 compares pixel selections on four examples from different benchmarks. Margin (Baseline) tends to select overlapping pixels mostly near object boundaries whereas our two-stage method evenly covers object boundaries, thin structures, and small classes – even under extremely low pixel budgets.

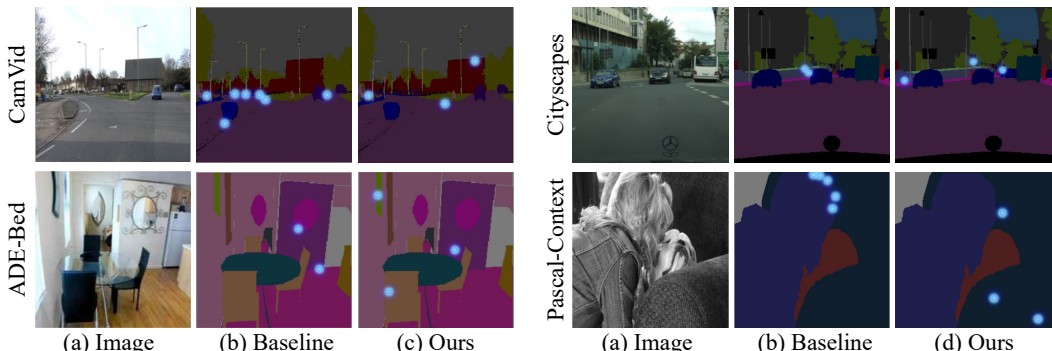

Figure 4: Qualitative pixel-selection comparison. (a) Input image; (b) baseline selections; (c) our selections. Light blue dots indicate selected pixels. Our method covers boundaries and fine details more broadly.

## 5 Conclusion

We addressed the problem of *low-budget active learning for semantic segmentation* by introducing a novel, two-stage pipeline that first narrows down candidate pixels through *clustering with diffusion-based representations*, then refines the selection via *disagreement-based uncertainty sampling*. Our approach effectively balances diversity and informativeness under extreme labeling constraints, as validated on four standard benchmarks. This work would open up a new direction for integrating advanced generative backbones such as diffusion models, to budget-constrained annotation pipelines, and paves the way for more practical, large-scale semantic segmentation systems.

**Limitations.** The proposed diffusion-based active learning by disagreement (DALD) is tailored for a diffusion-backbone; it requires sampling based on different random noises. Thus, DALD is applicable only with the diffusion-backbone. However, entropy-augmentation and two-stage sampling based on MaxHerding is agnostic to the choice of backbones.

## Acknowledgments and Disclosure of Funding

This work was supported by Institute of Information & communications Technology Planning & Evaluation (IITP) grant (No. RS-2022-00155966, Artificial Intelligence Convergence Innovation Human Resources Development (Ewha Womans University)) and the National Research Foundation of Korea (NRF) grant (No. RS-2025-16070597) funded by the Korea government (MSIT), as well as by the Natural Sciences and Engineering Research Council of Canada, the Canada CIFAR AI Chairs program, Calcul Québec, the BCI DRI Group, and the Digital Research Alliance of Canada.

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

# Appendix

## A  Broader Impacts

Our two-stage low-budget active learning pipeline can reduce annotation requirements by orders of magnitude, making high-quality semantic segmentation accessible in domains with scarce labeling resources – such as medical imaging (*e.g.*, histopathology, radiology), environmental monitoring (*e.g.*, land-cover mapping, wildlife surveys), and infrastructure inspection in developing regions. By building on a frozen, pre-trained diffusion backbone, we also cut down on repeated large-scale training, thereby lowering computational demands and carbon emissions. This democratizes cutting-edge segmentation tools for academic labs, start-ups, and NGOs that lack extensive annotation budgets or compute clusters.

On the other hand, any automated selection mechanism risks perpetuating biases present in the pre-trained diffusion model or in the small initial labeled set – potentially under-representing rare or sensitive classes (*e.g.*, skin lesions in medical scans, minority populations in urban scenes). Moreover, ease of pixel-level segmentation could be misused for large-scale surveillance or privacy-invasive monitoring if deployed without strict governance. We therefore recommend rigorous bias audits, transparent reporting of model behavior on under-represented groups, and adherence to data-privacy regulations when applying our approach in sensitive settings.

## B  Implementation Details

This section provides all settings needed to reproduce our experiments. All code, configuration files, and training/evaluation scripts are publicly available at `https://github.com/jn-kim/two-stage-edald`.

### B.1  Hyperparameters

Table 3 summarizes the key hyperparameters and implementation details used across the datasets in our experiments.

Table 3: Hyperparameters and implementation details across datasets.

| Details | ADE-Bed | CamVid | Cityscapes | Pascal VOC |
|---|---|---|---|---|
| Diffusion model | `lsun_bedroom` | `imagenet_256` | `imagenet_256` | `imagenet_256` |
| Resize method | Center crop | Bilinear | Center crop | Bilinear |
| Image resolution | | $256 \times 256$ | | |
| Diffusion Steps | | $\{50, 150, 250\}$ | | |
| Feature blocks | | $\{5, 8, 12, 17\}$ | | |
| Batch size | | 5 | | |
| Learning rate | | $1 \times 10^{-3}$ | | |
| Weight decay | | $1 \times 10^{-5}$ | | |
| Optimizer | | Adam | | |
| Scheduler | | cosineAnnealingLR | | |
| Classifier | | MLP | | |

**Notes:**

- `imagenet_256` refers to an unconditional ImageNet diffusion model trained at $256 \times 256$ resolution.
- `lsun_bedroom` refers to a diffusion model trained on LSUN bedroom data (three classes) at $256 \times 256$ resolution.
- Diffusion steps and feature blocks were chosen based on preliminary analysis to capture both low- and high-level semantics; see Section C.7 for further discussion.

## B.2  Feature Extraction from the Diffusion Model

To build comprehensive multi-scale representations, we extract intermediate features from a pre-trained diffusion model at denoising steps $t \in \{50, 150, 250\}$. These timesteps were selected from the later steps of the reverse diffusion process, as prior studies [15] have shown that activations at these stages yield more discriminative semantic features, thereby enhancing pixel-level predictions. For each timestep, we collect feature maps from selected decoder blocks (specifically, blocks 5, 8, 12, and 17). The exact shapes of the outputs of these blocks (excluding skip connections) during a forward pass are summarized in Table 4. These feature maps are first resized (using bilinear interpolation) to the input resolution ($256 \times 256$), and then concatenated channel-wise to form the final pixel-level representation.

Table 4: Output shapes of feature blocks extracted from the diffusion model. Highlighted rows indicate the selected blocks.

| Block Index | Output Shape | | Block Index | Output Shape |
|---|---|---|---|---|
| Block 0 | [1024, 8, 8] | | Block 9 | [512, 64, 64] |
| Block 1 | [1024, 8, 8] | | Block 10 | [512, 64, 64] |
| Block 2 | [1024, 16, 16] | | Block 11 | [512, 128, 128] |
| Block 3 | [1024, 16, 16] | | **Block 12** | **[256, 128, 128]** |
| Block 4 | [1024, 16, 16] | | Block 13 | [256, 128, 128] |
| **Block 5** | **[1024, 32, 32]** | | Block 14 | [256, 256, 256] |
| Block 6 | [512, 32, 32] | | Block 15 | [256, 256, 256] |
| Block 7 | [512, 32, 32] | | Block 16 | [256, 256, 256] |
| **Block 8** | **[512, 64, 64]** | | **Block 17** | **[256, 256, 256]** |

**Concatenation Strategy.** For each pixel, given $T = 3$ timesteps and $L = 4$ selected blocks per timestep, we resize and concatenate the corresponding feature maps to obtain a feature of shape:

$$\Big(T \times \big[\text{channels from Block 5 + Block 8 + Block 12 + Block 17}\big], \, 256, \, 256\Big).$$

For instance, if Block 5 outputs 1024 channels, Block 8 outputs 512 channels, and both Blocks 12 and 17 output 256 channels each, the final representation will have

$$3 \times (1024 + 512 + 256 + 256) = 3 \times 2048 = 6144 \quad \text{channels.}$$

## B.3  Structure and Training of the Segmentation Head

For pixel classification, we use a multilayer perceptron (MLP) with two hidden layers, incorporating ReLU activations and batch normalization layers, following the architecture used in [15]. The classifier takes a feature vector for each pixel as input and outputs a softmax probability distribution over the classes. Its parameters are updated using Adam optimizer with an initial learning rate of $1 \times 10^{-3}$ and a weight decay of $1 \times 10^{-5}$. We apply a cosine annealing learning rate scheduler with $T_{\max} = 5$ and $\eta_{\min} = 1 \times 10^{-6}$.

# C  Ablation Studies

## C.1  Hyperparameter Sensitivity

We study how our two-stage pipeline responds to the key hyperparameters $K$ (per-image representatives) and $M$ (global pool size) on CamVid. We vary

$$K \in \{20, 50, 100\}, \quad M \in \{0.25M_0, 0.4M_0, 0.5M_0\},$$

where $M_0 = N \times K$ is the size of the merged local pool (or initial global pool).

Figure 5 shows that our default hyperparameter ($K{=}100$, $M{=}0.5M_0$) yields 36.12 mIoU, while the best configuration ($K{=}20$, $M{=}0.4M_0$) achieves 38.70 mIoU. This indicates that reducing $K$ may not degrade – and can even improve – performance when paired with a suitably scaled global pool. Section C.8 discusses the resulting compute–accuracy trade-offs in more detail.

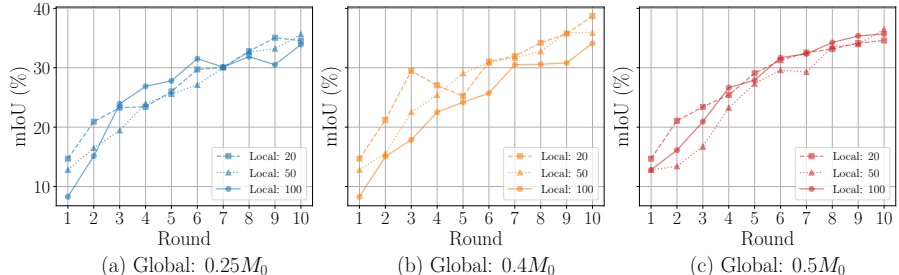

Figure 5: Hyperparameter sensitivity on CamVid: mIoU achieved after 10 AL rounds for varying per-image representative count $K \in \{20, 50, 100\}$ and global pool size $M \in \{0.25M_0, 0.4M_0, 0.5M_0\}$. Each subplot shows one $M$ setting.

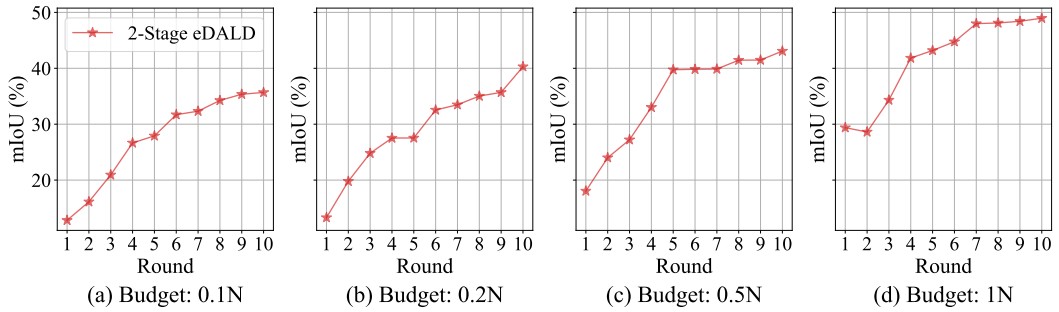

Figure 6: Results on CamVid with various budgets: $0.1N$, $0.2N$, $0.5N$, and $1N$ pixels per round.

## C.2   Budget Sensitivity

Figure 6 plots the mIoU of our two-stage eDALD over 10 AL rounds on CamVid under four annotation budgets: $0.1N$, $0.2N$, $0.5N$, and $1N$ pixels per round (with $0.1N \approx 0.0015\%$ of all pixels). Under the smallest budget ($0.1N$), two-stage eDALD converges to $\sim 35.7\%$ mIoU but improves more rapidly in early rounds. As the budget increases, the final (round 10) mIoU of eDALD rises accordingly: $40.3\%$ at $0.2N$, $43.1\%$ at $0.5N$, and $48.9\%$ at $1N$.

## C.3   Convergence to Fully Supervised: Additional Details

This section expands on the convergence analysis discussed in Section 4.3. Table 5 reports, for each dataset, the number of rounds and labeled-pixel fractions required to reach $80\%$ and $90\%$ of the fully supervised mIoU under budget $b = 0.1N$ (*i.e.*, on average 0.1 pixel/image/round). We use two-stage eDALD for rounds $1-10$ and switch to Margin thereafter.

Across datasets, two-stage eDALD (rounds 1–10) followed by Margin converges to strong performance with *vanishingly few* labeled pixels. Reaching $80\%$ of fully supervised mIoU takes only 11–27 rounds, corresponding to 0.0018–0.0043% of all pixels. For $90\%$, convergence occurs within 21–47 rounds, using merely 0.0034–0.0073% of pixels ($\approx$ one out of 30k to 14k pixels). Dataset-

Table 5: Rounds and labeled-pixel fraction (percentage of all pixels) needed to reach 80% and 90% of the fully supervised mIoU (FSL) under the $b = 0.1N$ budget. We use two-stage eDALD for rounds $1-10$ and switch to Margin thereafter.

| Target | Metric | CamVid | ADE-Bed | Cityscapes | Pascal-C |
|--------|--------|--------|---------|-----------|----------|
| 80% FSL | Rounds | 19 | 27 | 12 | 11 |
|         | %Pixels | 0.0031% | 0.0043% | 0.0020% | 0.0018% |
| 90% FSL | Rounds | 32 | 47 | 28 | 21 |
|         | %Pixels | 0.0050% | 0.0073% | 0.0044% | 0.0034% |

wise, Pascal-Context is fastest to 90% (21 rounds, 0.0034%), followed by Cityscapes (28, 0.0044%) and CamVid (32, 0.0050%), while ADE-Bed is most demanding (47, 0.0073%), likely reflecting indoor-scene variability and pretraining mismatch. These results substantiate our switching policy: diversity → uncertainty (two-stage) is most sample-efficient under extreme budgets, and a late switch to pure uncertainty (Margin) accelerates the final approach to the supervised ceiling. Overall, near-supervised quality is attainable at labeling rates that are orders of magnitude below $1\%$, reinforcing the practicality of our low-budget AL setting.

## C.4 Additional Backbone Results

To compare our DDPM-based two-stage eDALD with other architectures, we extend the two-stage framework to two widely used backbones, DeepLabV3 [52] and ViT [3]. Since eDALD leverages the intrinsic stochasticity of diffusion models, we employ its counterpart, eBALD, which estimates entropy-augmented mutual information via Monte Carlo dropout for non-diffusion backbones. The results in Table 6 demonstrate that the DDPM-based approach consistently outperforms the others across all datasets, highlighting the strong synergy between diffusion-driven uncertainty estimation and MaxHerding under the low-budget regime.

We further present a ViT-specific comparison in Table 7, comparing the two-stage eBALD with single-stage Margin and BalEntAcq. The two-stage approach consistently achieves the highest final-round mIoU, demonstrating that entropy-augmented disagreement remains beneficial even for transformer-based backbones.

Table 6: mIoU (%) of two-stage methods under a low-budget regime (10 rounds) with three backbones: DeepLabV3, ViT, and DDPM. DDPM uses 2-Stage eDALD, while DeepLabV3 and ViT use 2-Stage eBALD.

| Backbone | CamVid | ADE-Bed | Cityscapes | Pascal-C |
|---|---|---|---|---|
| DeepLabV3 [52] | 29.48 | 9.75 | 31.64 | 28.52 |
| ViT [3] | 31.48 | 10.80 | 32.27 | 31.98 |
| DDPM [13] | **36.12** | **31.12** | **33.34** | **47.98** |

Table 7: ViT-specific comparison of active learning methods in mIoU (%) under the low-budget regime.

| Method | CamVid | ADE-Bed | Cityscapes | Pascal-C |
|---|---|---|---|---|
| Margin | 29.52 | 9.51 | 27.36 | 29.43 |
| BalEntAcq | 24.67 | 9.81 | 21.05 | 30.53 |
| 2-Stage eBALD | **31.48** | **10.80** | **32.27** | **31.98** |

## C.5 Comparison with Diversity-based Methods

To assess the effectiveness of different diversity-driven selection strategies, we compare pipelines using either MaxHerding [16] or Core-set [9] as the first-stage selector, followed by eDALD for uncertainty-based refinement. Core-set corresponds to the $k$-center greedy algorithm, which iteratively selects the farthest sample in feature space to ensure coverage. Table 8 summarizes the final-round mIoU across four datasets, comparing both one-stage and two-stage variants with two backbones: DeepLabV3 and DDPM.

The results reveal two consistent trends. First, MaxHerding provides stronger coverage than Core-set across all configurations, whether used alone or within a two-stage pipeline. Second, this advantage is consistent across both backbones but is particularly pronounced for DDPM, where the combination MaxHerding → eDALD achieves the best overall performance. An exception is observed on Pascal-Context, where MaxHerding alone achieves higher performance than its two-stage variant. We attribute this to the dataset's large class count (33) and diverse scene distribution, where maximizing pure feature-space coverage can be more effective for capturing rare or spatially scattered classes. Overall, these findings demonstrate that while MaxHerding alone is a strong representation-based baseline, its combination with eDALD uncertainty consistently yields superior performance in most practical scenarios.

Table 8: Final-round mIoU comparison of diversity methods (Core-set *vs*. MaxHerding) under the low-budget regime. For DeepLab we report two-stage variants with eBALD, while for DDPM we report both one-stage and two-stage with eDALD.

| Backbone | Method | ADE-Bed | CamVid | Cityscapes | Pascal-C |
|---|---|---|---|---|---|
| DeepLabV3 [52] | Core-set | 9.21 | 26.49 | 30.42 | 25.83 |
| | MaxHerding | 9.64 | 28.78 | 31.05 | 28.12 |
| | Core-set $\rightarrow$ eBALD | 9.53 | 26.50 | 30.48 | 26.70 |
| | MaxHerding $\rightarrow$ eBALD | 9.70 | 29.24 | 31.45 | 28.50 |
| DDPM [13] | Core-set | 18.49 | 16.19 | 22.74 | 23.27 |
| | MaxHerding | 24.06 | 31.83 | 25.80 | **52.04** |
| | Core-set $\rightarrow$ eDALD | 26.70 | 32.95 | 32.20 | 44.58 |
| | MaxHerding $\rightarrow$ eDALD | **31.12** | **36.12** | **33.34** | 47.98 |

## C.6 Pixel-to-Region Expansion with SAM

We extend our two-stage eDALD pipeline to practical *region*-level supervision by integrating SAM [53] under a strict *click-parity* budget ($b=0.1N$, *i.e.*, on average 0.1 pixel/image/round). Each selected pixel is used as *one* positive point prompt for SAM (no extra clicks and no iterative refinements). SAM returns multiple mask proposals with predicted IoU scores; we keep the highest-scoring proposal if it exceeds a confidence threshold $\tau$ and otherwise fall back to the single-pixel label. Thus, each click can expand to a dense region label when reliable, yielding many more supervised pixels without increasing annotation cost.

Table 9 reports 10-round results for the baseline two-stage eDALD (w/o SAM) and the SAM-augmented variant (w/ SAM). Across datasets, SAM consistently *accelerates early rounds* and often improves final mIoU under the same budget: on CamVid the final gain is $+5.24$ pp (41.34 *vs*. 36.10), while ADE-Bed, Cityscapes, and Pascal-Context show modest but steady improvements ($+0.48$ pp, $+0.68$ pp, and $+0.27$ pp, respectively). Overall, single-click, region-level supervision via SAM provides faster convergence and better cost-efficiency in the extreme low-budget setting.

Table 9: Active learning performance (mIoU, %) over 10 rounds for two-stage eDALD with and without SAM under the $b=0.1N$ budget (*strict click parity*: one positive point per selected pixel).

| Dataset | Method | R1 | R2 | R3 | R4 | R5 | R6 | R7 | R8 | R9 | R10 |
|---|---|---|---|---|---|---|---|---|---|---|---|
| CamVid | w/o SAM | 17.90 | 18.63 | 24.52 | 27.00 | 28.40 | 27.40 | 29.50 | 32.90 | 34.60 | 36.10 |
| | w/ SAM | 20.51 | 28.68 | 32.03 | 28.74 | 31.93 | 34.78 | 37.75 | 36.74 | 39.56 | 41.34 |
| ADE-Bed | w/o SAM | 12.70 | 15.40 | 18.70 | 22.60 | 25.30 | 27.10 | 28.20 | 28.50 | 30.20 | 31.20 |
| | w/ SAM | 14.00 | 16.37 | 19.94 | 24.13 | 29.11 | 29.58 | 30.34 | 31.49 | 32.43 | 31.68 |
| Cityscapes | w/o SAM | 20.40 | 24.20 | 26.57 | 29.06 | 30.01 | 30.40 | 31.00 | 31.80 | 31.90 | 33.34 |
| | w/ SAM | 20.31 | 25.85 | 27.20 | 26.37 | 26.55 | 27.20 | 28.05 | 29.83 | 33.89 | 34.02 |
| Pascal-Context | w/o SAM | 25.80 | 29.75 | 36.61 | 39.89 | 41.42 | 43.43 | 44.33 | 45.98 | 46.24 | 47.44 |
| | w/ SAM | 30.42 | 38.98 | 40.45 | 41.90 | 42.33 | 43.22 | 43.48 | 44.53 | 46.87 | 47.71 |

## C.7 Decoder-Block Selection Ablation

We compare four decoder-block configurations for multi-scale feature extraction on ADE-Bed:

- $\{2,3\}$: deep blocks (high-dim features, low spatial resolution)
- $\{11\text{–}17\}$: shallow blocks (low-dim features, high spatial resolution)
- $\{5,6,7,8,12\}$: original LEDM setting [15]
- $\{5,8,12,17\}$: our compact selection

Table 10 reports mIoU for each. The compact set $\{5,8,12,17\}$ (45.58%) matches the original five-block configuration (46.41%), while deep-only $\{2,3\}$ (42.19%) and shallow-only $\{11\text{–}17\}$ (22.68%) both degrade significantly. This confirms that retaining both spatial detail and high-level semantics is crucial. We therefore adopt $\{5,8,12,17\}$ in all experiments – it reduces feature dimensionality by $\sim 28\%$ and speeds up without sacrificing accuracy.

Table 10: Ablation on decoder-block choice (ADE-Bed). Channel dim. is the sum over selected blocks.

| Blocks | Channel Dim. | mIoU (%) |
|---|---|---|
| $\{2, 3\}$ (deep-only) | 6,144 | 42.19 |
| $\{11-17\}$ (shallow-only) | 6,144 | 22.68 |
| $\{5, 6, 7, 8, 12\}$ (original) | 8,448 | 46.41 |
| $\{5, 8, 12, 17\}$ (compact) | 6,144 | 45.58 |

## C.8 Computational Complexity

The dominant cost in our pipeline is candidate selection via MaxHerding. Given a global pool of size $M$ with $d$-dimensional features and per–round budget $B$, forming the full RBF kernel costs $\mathcal{O}(M^2 d)$ time and $\mathcal{O}(M^2)$ memory. The subsequent greedy selection over $B$ points adds $\mathcal{O}(M^2 B)$ time, giving an overall complexity of $\mathcal{O}\big(M^2 \cdot \max\{d, B\}\big)$. Since $M = N \times K$ grows with the number of images $N$ and the per–image candidate count $K$, this step can become the bottleneck on large datasets.

**Runtime/memory in practice.** On ADE-Bed, our two-stage eDALD with $N$=964, $K$=50 ($M$=48,200) required: feature extraction (8m 22s, 25.4 GB), local herding (4m 24s, 3.7 GB), and global herding (14m 27s, 38.1 GB), compared to training (19m 14s, 22.7 GB). Overall, MaxHerding accounts for $\sim 41\%$ of the per-round wall-clock time, and global herding dominates peak memory (38.1 GB).

**Reducing peak memory.** To further lower the memory footprint, we adopt a split-and-herd variant: partition the global pool into memory-safe splits, allocate a proportional sub-budget to each split, and run greedy herding sequentially while conditioning on selections from earlier splits to preserve cross-split coverage. This relaxes full optimality but prevents out-of-memory under higher $K$ or high-resolution settings, effectively reducing *peak* memory from $\mathcal{O}(M^2)$ to roughly $\mathcal{O}\big((M/S)^2\big)$ per split when using $S$ splits, with negligible impact on accuracy in our experiments.

# D Qualitative Results

We qualitatively compare some active learning selection results from the proposed two-stage eDALD against the single-stage Margin baseline on each of four datasets.

## D.1 Final Pixel Selections

Figure 7 overlays the $b$ selected pixels on the ground-truth maps for both methods. Margin tends to concentrate selections on boundary regions and often chooses redundant pixels within the same class region, resulting in less diverse annotations. In contrast, our two-stage pipeline disperses pixels evenly across object edges, small classes, and scene context, ensuring that the most informative locations are annotated.

## D.2 Selection Progression

Figure 8 visualizes the selected pixels of two-stage eDALD across rounds. Early rounds focus on broad structural cues; later rounds refine boundary regions and rare classes. This progression highlights how representation filtering first ensures coverage, then eDALD uncertainty hones in on the remaining ambiguous pixels.

## D.3 Segmentation Outputs

For representative images, Figure 9 shows the predicted segmentation maps. Margin often misses thin structures and small objects (*e.g.*, distant pedestrians, traffic signs), producing fragmented or smoothed regions. In contrast, our eDALD-trained model yields cleaner boundaries and recovers fine details more faithfully.

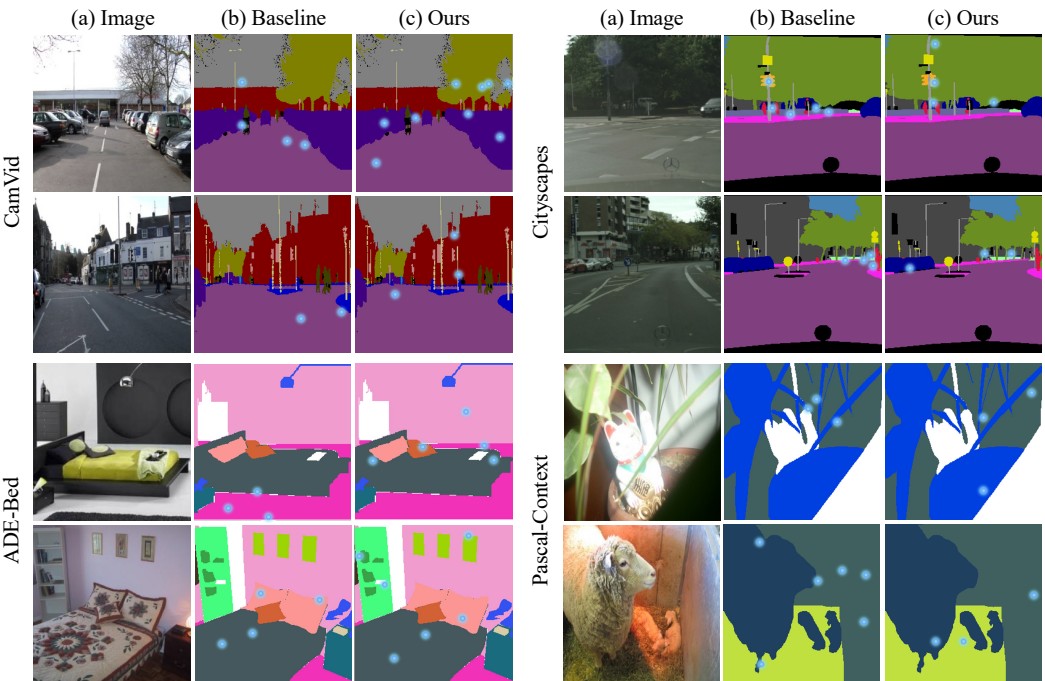

Figure 7: Final pixel selections overlaid on ground truth: (a) input image; (b) Margin selections; (c) two-stage eDALD selections. Light blue dots mark chosen pixels.

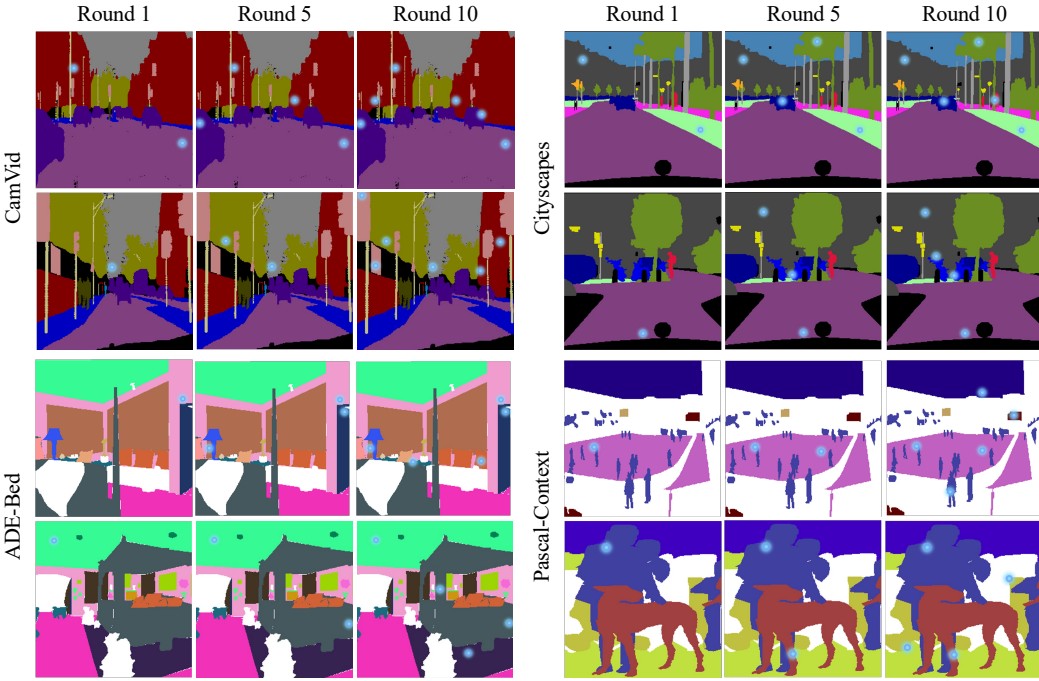

Figure 8: Two-stage eDALD pixel selections at rounds 1, 5, and 10 on multiple example images.

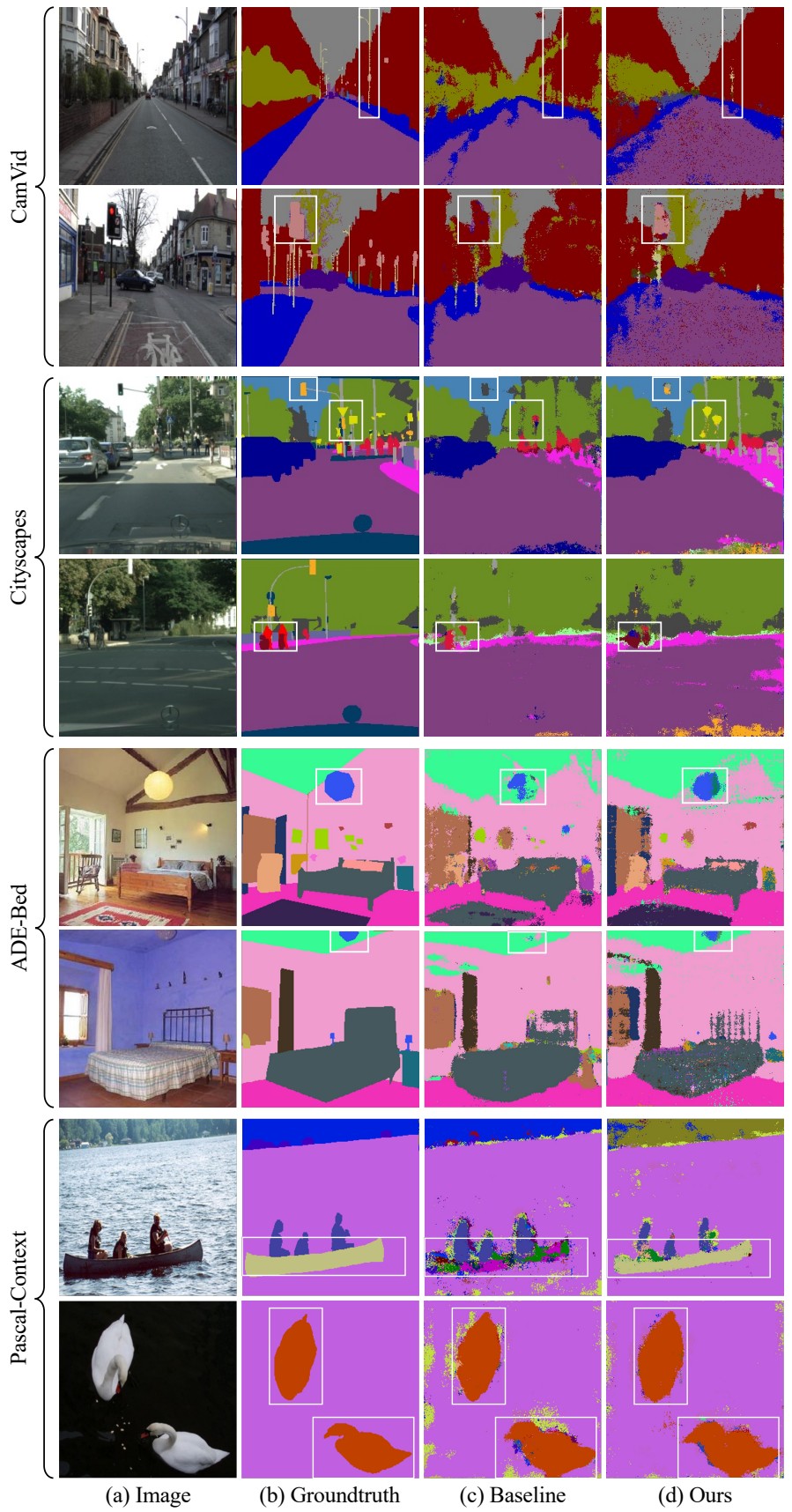

Figure 9: Example segmentation results. Columns show (a) input image, (b) ground truth, (c) Margin prediction, and (d) 2-stage eDALD prediction.

