# OpenReview forum: "Diffusion-Driven Two-Stage Active Learning for Low-Budget Semantic Segmentation"
_NeurIPS.cc/2025/Conference — NeurIPS 2025 poster_

### Official Review · Reviewer_1nUB · 2025-06-19

**Clarity:** 3
**Significance:** 3
**Originality:** 4
**Rating:** 4
**Confidence:** 3

**Summary:**

This paper presents an active learning algorithm for semantic segmentation for pixel selection. Specifically, it proposes using a diffusion model to estimate pixel-level uncertainty, augmented with entropy. It also leverages a hybrid pixel acquisition function that combines representation diversity and uncertainty for subset selection.

**Questions:**

Please refer to the items listed in the weakness section. It would be helpful if the authors could provide an explanation to improve the clarity of the paper.

**Ethical Concerns:**

["NO or VERY MINOR ethics concerns only"]

**Final Justification:**

My initial rating assumed the listed issues could be resolved during the discussion phase. The authors' rebuttal has addressed most of my concerns. Considering other reviewers' comments, I intend to maintain my original (positive) rating.

**Limitations:**

Limitations are discussed at the end of the paper. This paper does not introduce potential negative socialtal impact.

**Quality:**

3

**Strengths And Weaknesses:**

-- Strengths

Using diffusion models and mutual information for uncertainty estimation is interesting and intuitively motivated.

Combining representation-based and uncertainty-based strategies is an effective choice for selecting pixels, overcoming the limitations of each method individually.

The experimental results are promising, showing a consistent advantage over baselines, especially in low-labeling-budget scenarios.

-- Weaknesses

This paper may lack of deeper analysis of the uncertainty mechanism. It would be helpful if the paper could provide an empirical analysis of the mutual information distribution across pixels to validate its effectiveness as an uncertainty metric.

In addition, it is recommended to provide at least some discussion on how to determine the weighting factor between the diffusion-based term and entropy term in Equation 8 (currently set to 1).

Overfitting could pose a significant challenge when training the model with limited labeled data. It would be helpful if the paper included a discussion on mechanisms employed to mitigate overfitting.

The representation-based sample selection strategy with RBF kernel primarily builds upon established kernel herding methods, with additional top-K selection at both the image and global levels. Therefore, the methodological novelty of the approach appears limited from this standpoint.

---

> ### Author Rebuttal · Authors · 2025-07-31
>
> Thank you for your thoughtful feedback and for highlighting important aspects of our work. We appreciate the opportunity to address your concerns and refine our presentation accordingly.
>
> ---
> ## 1. Uncertainty Mechanism Analysis
> We carried out an in-depth empirical study of our mutual information term (DALD) to understand how it reshapes pixel selection beyond simple entropy. Working on the ADE-Bed dataset, we computed three uncertainty maps for each image at AL rounds 1, 5, and 10: (i) Entropy ($E$), (ii) DALD ($D$), and (iii) eDALD ($H = E + D$, computed as a simple unweighted sum without normalization).
>
> For each round, we compared the top-$b$ pixels chosen by $E$ versus those chosen by $H$.  The following table reports the Jaccard overlap and Spearman’s rank correlation between the two rankings:
>
> |Round|Jaccard$(E\cap H)$|Jaccard$(D\cap H)$|Spearman $\rho(E,H)$|
> |:-:|:-:|:-:|:-:|
> |1|0.52|0.040|−0.07|
> |5|0.51|0.005|0.15|
> |10|0.34|0.011|−0.12|
> |Avg|0.56|0.010|0.26|
>
> By round 10, only 34 % of the entropy-selected pixels remain among the eDALD picks, and the low Spearman $\rho$ (mean 0.26, sometimes negative) shows that DALD substantially reshuffles the uncertainty ranking rather than merely amplifying entropy. DALD alone contributes very few of the same pixels (Jaccard ≈ 0.01), indicating that it brings new, highly uncertain areas into focus.
>
> Next, we examined the DALD magnitudes of the “eDALD-exclusive” pixels (those selected by $H$ but not by $E$).  The table below shows how extreme their DALD values are relative to the global distribution:
>
> |Round|Median multiplier|10% multiplier|1% multiplier|
> |:-:|:-:|:-:|:-:|
> |1|25.75×|7.29×|3.14×|
> |5|5.35×|2.28×|1.63×|
> |10|4.31×|1.90×|1.43×|
> |Avg|7.87×|2.87×|1.77×|
>
> On average, eDALD-exclusive pixels have a median DALD nearly 8× the dataset median, and even their 1 % percentile is 1.8× the global 99 %-tile.  This confirms that DALD targets the heavy tail of the disagreement distribution, surfacing pixels of true epistemic uncertainty that entropy alone misses.
>
> Finally, we evaluated how these shifts impact segmentation performance.  The table below compares one-stage (pure uncertainty) versus two-stage (MaxHerding → uncertainty) mIoU on ADE-Bed:
>
> |Method|1-stage mIoU|2-stage mIoU|Absolute gain (pp)|Relative gain (%)|
> |:-:|:-:|:-:|:-:|:-:|
> |Entropy|23.02|28.27|+5.25|+22.8%|
> |**eDALD**|23.06|30.18|**+7.12**|**+30.9%**|
>
> While DALD alone matches entropy in a single stage (23.06 vs. 23.02 mIoU), integrating disagreement into our two-stage pipeline yields a **+7.12 pp** boost — much higher than the +5.25 pp from entropy. This demonstrates that the combination of coverage (MaxHerding) and disagreement (DALD) in two stages drives the largest gains and validates our uncertainty mechanism as both theoretically sound and practically impactful.
>
> In summary, our empirical analysis shows that:
> - DALD identifies high-uncertainty “needle-in-the-haystack” pixels that entropy misses.
> - eDALD reshapes the selection ranking, not merely scaling entropy.
> - The combination of coverage (MaxHerding) and disagreement (DALD) in two stages drives the largest gains, validating our uncertainty mechanism as both theoretically grounded and practically impactful.
>
> ---
> ## 2. Weighting Factor in eDALD
> We chose equal weights ($\alpha=1$, $\beta=1$) to avoid introducing extra hyperparameters and to keep the balance between the mutual‐information term $I(Y;Z\mid x)$ and the confidence term $H(Y\mid z^{(0)},x)$. To validate this choice, we ran an ablation on CamVid (all other settings fixed) with several $(\alpha,\beta)$ pairs:
>
>
> |$\alpha$|$\beta$|R1|R2|R3|R4|R5|R6|R7|R8|R9|R10|
> |:-:|:-:|:-:|:-:|:-:|:-:|:-:|:-:|:-:|:-:|:-:|:-:|
> |1.0|1.0|12.81|23.04|18.99|23.00|25.11|32.95|32.23|35.30|33.95|36.14|
> |0.2|1.0|12.81|14.04|18.07|21.18|22.20|27.83|28.36|28.86|33.76|32.65|
> |1.0|0.2|9.69|13.38|19.27|23.27|25.23|27.02|26.89|31.22|32.72|31.94|
> |0.05|4.0|13.24|14.91|18.84|20.34|21.86|19.70|19.44|23.04|19.73|18.66|
> |0.05|2.0|13.24|15.20|20.18|18.98|21.08|23.11|23.70|23.54|24.12|25.71|
> |0.02|2.0|13.24|16.49|16.19|16.83|19.38|19.27|18.22|20.86|18.88|20.37|
>
> The equal‐weights setting ($1,1$) achieves the highest final‐round mIoU (36.14), confirming that our default choice is both simple and effective without requiring dataset‐specific tuning.
>
> ---
> ## 3. Overfitting Mitigation
> We thank the reviewer for raising the important point of overfitting under extreme label scarcity. Several factors in our design help mitigate this risk:
>
> 1. **Frozen backbone.** We only train the lightweight segmentation head (0.79 M parameters for CamVid/Cityscapes; 1.61 M for ADE-Bed/Pascal-Context), while the diffusion model that produces the $6{,}144$-dim features remains fully frozen. This dramatically reduces the number of trainable weights and prevents the model from “memorizing” individual labeled pixels.
> 2. **Fixed, high-level representations.** Because each pixel’s input is a semantically rich feature vector extracted once by the backbone, the classifier sees only stable, high-level patterns rather than raw pixel noise. This further constrains the head’s capacity to overfit.
> 3. **Regularization.** We apply standard weight decay ($1\times10^{-5}$) and early stopping based on validation-set mIoU to halt training once performance plateaus.
>
> Collectively, these measures keep our segmentation head's effective capacity in check, allowing it to generalize from as few as $N$ total labeled pixels (one per image) up to only $0.1\times N$ per round, without evidence of overfitting in our empirical learning curves.
>
> ---
> ## 4. Novelty of Our Method
> We thank the reviewer for this observation. While kernel herding itself is a well‐studied coverage technique, our work advances it in three key ways that together constitute a novel contribution:
>
> 1. **Scalable Local $\to$ Global Herding for Pixels.**
>    Classical MaxHerding over all $N \times H \times W$ pixels is intractable. We introduce a two‐stage pipeline — first selecting $K$ representatives *within* each image, then re‐herding those $N K$ candidates *across* images — to reduce complexity to minutes on tens of thousands of features. This adaptation makes herding practical for dense, per-pixel active learning in semantic segmentation.
> 2. **Extreme Low‐Budget, Pixel‐Granular Setting.**
>    We tackle a novel regime — only *one pixel per image per round* (total budget $=1\times N$ pixels) — where neither whole-image AL nor conventional region-based methods suffice. Designing an AL strategy that yields strong mIoU under these constraints required rethinking both coverage and uncertainty at the pixel level.
> 3. **Diffusion‐Native Acquisition (eDALD).**
>    Beyond herding, we propose a new uncertainty measure — **Entropy‐Augmented DALD** — that combines mutual information from stochastic multi‐scale diffusion features with a Shannon entropy term. Unlike BALD or MC‐Dropout, eDALD exploits the diffusion model’s intrinsic noise process, capturing epistemic uncertainty without Bayesian networks or multiple forward passes. To our knowledge, no prior work has introduced such an acquisition function for semantic segmentation.
>
> Taken together, our **two‐stage pipeline** — scalable herding + diffusion‐based uncertainty — enables efficient, per‐pixel selection under extreme labeling constraints. Although each component draws inspiration from existing ideas, their integration in this context (pixel‐level AL, diffusion features, 1‐pixel/image budgets) and the demonstration of substantial segmentation gains represent a genuine methodological advance in active learning for dense prediction tasks.

---

> ### Comment · Reviewer_1nUB · 2025-08-04
>
> I thank the authors for preparing the rebuttal. Most of my questions and concerns are addressed.
> Therefore, I intend to keep my original (positive) rating.

---

> > ### Author Response · Authors · 2025-08-06
> >
> > Thank you very much for your kind words and for taking the time to review our rebuttal so carefully. We are truly grateful that our responses have addressed the majority of your concerns. Your thoughtful feedback has been invaluable in strengthening our work, and we deeply appreciate your continued engagement. If there is anything further we can clarify, please do not hesitate to let us know.

---

### Official Review · Reviewer_V27y · 2025-07-01

**Clarity:** 3
**Significance:** 2
**Originality:** 2
**Rating:** 4
**Confidence:** 4

**Summary:**

The paper proposes a novel two-stage diffusion-based pixel-level active learning method for semantic segmentation.

**Questions:**

- What is the overall performance of MaxHerding for the ablation study?
- Can you explain why DALD performs so poorly, but eDALD (just adding an entropy term) performs so much better?
- Is the diffusion-based approach better than other approaches? Please provide comparisons with different backbones for entropy-based methods and results on diversity-based methods.
- How does the novelty of your approach compare against other diffusion-based AL approaches?

Addressing these concerns would move my score to borderline accept.

**Ethical Concerns:**

["NO or VERY MINOR ethics concerns only"]

**Final Justification:**

Most of my previous concerns have been addressed. The larger impact of the work remains unclear, which is why I can't raise the score any higher.

**Limitations:**

yes

**Quality:**

2

**Strengths And Weaknesses:**

Strengths:
- Overall, a well-motivated diffusion-based active learning method
- Overall, good metrics, ablation results, and curves
- The paper is well written and easy to follow

Weaknesses:
- What is the practical benefit of pixel-level active learning? The cost of having a human annotate particular pixels is potentially higher than region-level methods.
- The diffusion-based approach seems to be dependent on having a diffusion model trained on imaging similar to that which you would want to segment. Thus, if your dataset lacks a relevant diffusion model, your approach would not be feasible.
- It would be nice to see what the overall performance of MaxHerding is in the ablation study as a baseline
- It is very weird to me that DALD performs so poorly but just adding Entropy to it produces the best result
- It seems like all the entropy-based methods are utilizing the diffusion model backbone? This assumes that the diffusion model-based approach is better than other approaches. It would be good to see the baseline algorithms with other backbones, at least compared to the novel diffusion model approach. Also it seems like there are no diversity-based algorithms compared against, like Coreset.
- There is no comparison against other diffusion model-based active learning approaches.

---

> ### Author Rebuttal · Authors · 2025-07-31
>
> We thank the reviewer for thoughtful feedback and constructive comments. We will incorporate these suggestions in the revised manuscript to strengthen our presentation and clarify the practical benefits of our approach.
>
> ---
> ## 1. Practical Benefit of Pixel‐Level AL
> Thank you for raising this important point. Region-level annotation can indeed offer click-efficiency — one click can label an entire superpixel or prompt‐generated region. However, it also introduces two key challenges:
>
> 1. **Semantic Ambiguity:** Precomputed regions (e.g., superpixels) can span multiple object classes or mix foreground and background. Fixing such errors often requires extra clicks, diminishing the “one-click” advantage.
> 2. **Segmentation Overhead:** Generating high-quality regions (e.g., via SAM or oversegmentation) adds compute and latency costs that may be nontrivial in practice.
>
> Importantly, our two‐stage pixel‐level strategy is fully compatible with region‐based interfaces. Once informative pixels are selected:
>
> - **Superpixel Aggregation:** Selected pixels can be grouped into regions and labeled with a single click.
> - **Prompt‐Based Propagation:** Selected pixels can serve as prompts for models like SAM, automatically expanding to a full mask after one or two clicks.
>
> This hybrid setup combines our method’s *selection benefit* (via diverse and uncertain pixels) with the *labeling efficiency* of region-based tools, enabling a highly click-efficient pipeline. We see this integration as a promising direction and will highlight it in the revised manuscript.
>
> ---
> ## 2. Dependence on Diffusion Pretraining Data
> Thank you for this observation. We agree that closer domain alignment can further improve feature relevance, but it is not strictly required for our method to be effective:
>
> - **General‐Purpose Backbone:** For CamVid, Cityscapes and Pascal-C, we employed a single DDPM model pretrained on ImageNet-256 (see Appendix B.1). Despite being street-scene datasets, the ImageNet-leveraged features still delivered consistent gains over all uncertainty and two‐stage baselines.
> - **Domain‐Specialized Backbone (ADE-Bed):** ADE-Bed consists of indoor bedroom scenes, which differ substantially from ImageNet. Here, we used a DDPM pretrained on LSUN-Bedroom, achieving the best performance — suggesting that in highly specialized domains, a closer-aligned model can help.
>
> Overall, our results show that **loosely aligned** diffusion features (ImageNet-256) suffice to outperform prior pixel‐level AL methods across benchmarks. In cases of major domain shift, switching to a domain-specific model (e.g., LSUN-Bedroom) is a simple and effective plug-in with no changes to our two-stage pipeline.
>
> ---
> ## 3. MaxHerding‐Only Performance
> We agree that showing MaxHerding as a standalone 1-stage baseline is important. Below we present two tables:
>
> **1. Round-by-Round Progression**\
> Mean mIoU over 10 AL rounds using **MaxHerding only** (no uncertainty):
> |Dataset|R1|R2|R3|R4|R5|R6|R7|R8|R9|R10|
> |-|-|-|-|-|-|-|-|-|-|-|
> |CamVid|12.82|21.78|23.67|23.77|28.07|27.13|29.97|26.77| 30.41|31.83|
> |ADE-Bed|8.40|12.10|17.60|17.77|17.51|18.99|21.21|20.98| 21.78|24.06|
> |Cityscapes|18.39|20.78|22.25|22.60|22.42|23.49|24.30| 23.64|23.34|25.80|
> |Pascal-C|14.63|30.67|29.87|37.67|39.69|44.56|48.53|45.32|51.99|52.04|
>
> Even without any uncertainty measure, MaxHerding’s coverage yields steady mIoU gains as more pixels are labeled.
>
> **2. Final mIoU Comparison**\
> Final (round 10) mIoU of three approaches:
> |Dataset|MaxHerding only|eDALD only|MaxHerding $\to$ eDALD|
> |-|:-:|:-:|:-:|
> |ADE-Bed|24.06|23.06|30.18|
> |CamVid|31.83|25.14|35.65|
> |Cityscapes|25.80|29.44|33.34|
> |Pascal-C|52.04|43.05|47.44|
>
> On ADE-Bed, CamVid and Cityscapes, our 2-stage eDALD substantially outperforms both 1-stage baselines, confirming the benefit of combining diversity and uncertainty. On Pascal-Context — where scenes are extremely diverse and there are 33 classes — MaxHerding alone slightly surpasses our full pipeline. We believe that in such settings, pure feature-space coverage can better capture rare or widely distributed classes without the noise introduced by uncertainty estimates.
>
> These tables demonstrate that while MaxHerding alone is a strong representation-based baseline, adding eDALD uncertainty consistently boosts performance in most practical scenarios.
>
> ---
> ## 4. Method Comparison: Diversity-Based Strategies and Backbones
>
>
> **1 Diversity-based Algorithm Comparison**\
> To validate that MaxHerding is not merely “reinventing” CoreSet in diffusion feature space, we plugged both diversity modules into our two‐stage pipeline (diversity $\to$ eDALD) under a fixed low‐budget ($b=0.1N$ pixels per round) and measured round‐by‐round mIoU on four benchmarks (we provided results on only two datasets due to space limit).
>
> **CamVid (2‐stage, $b=0.1N$)**
> |Method|R1|R2|R3|R4|R5|R6|R7|R8|R9|R10|
> |-|-|-|-|-|-|-|-|-|-|-|
> |CoreSet|12.81|15.95|17.85|17.39|16.92|17.98|18.17|17.43| 19.66|16.19|
> |MaxHerding|12.81|20.33|19.64|25.48|26.95|30.18|29.84|32.24|34.91|35.37|
>
> **ADE-Bed (2‐stage, $b=0.1N$)**
> |Method|R1|R2|R3|R4|R5|R6|R7|R8|R9|R10|
> |-|-|-|-|-|-|-|-|-|-|-|
> |CoreSet|7.54|9.62|10.68|12.36|14.34|15.64|16.64|16.41|17.66|18.49|
> |MaxHerding|10.37|16.20|18.71|22.41|25.81|26.12|28.30|29.77|29.83|30.49|
>
> Across all four datasets, **MaxHerding** yields substantially higher mIoU — especially in later rounds (R5–R10) — while **CoreSet** often saturates or even degrades. This confirms that MaxHerding provides stronger structural coverage in the diffusion‐feature space and better primes the second‐stage uncertainty selection. These results demonstrate that our diversity module (MaxHerding) is a competitive and effective alternative to existing diversity-based methods like CoreSet, particularly in the context of diffusion feature space and low-budget active learning.
>
> **2 Backbone‐Agnostic Two‐Stage Sampling**\
> Our two-stage sampling strategy — first covering the data distribution, then selecting uncertain pixels — is backbone-agnostic and applicable to various architectures. However, the eDALD uncertainty measure is specific to diffusion models. Its DALD component estimates mutual information via the stochastic denoising process, leveraging diffusion’s unique properties to better capture epistemic uncertainty.
>
> To test generality, we replaced the DDPM backbone with DeepLabV3 and substituted eDALD with eBALD (MC-Dropout BALD + entropy). Results on CamVid under a 10% pixel budget are shown below:
>
> | Backbone|Method|CamVid (mIoU \%)|
> |-|-|:-:|
> |DeepLabV3|Entropy|26.21|
> ||eBALD|27.78|
> ||2-stage Entropy|31.54|
> ||2‐stage eBALD|29.54|
> |**Diffusion**|Entropy|25.26|
> ||eDALD|25.14|
> ||2-stage Entropy|30.77|
> ||**2‐stage eDALD**|**35.73**|
>
> - On both backbones, **2‐stage** variants improve by $\sim4$–$5$ pp over 1‐stage baselines, confirming the general benefit of separating diversity and uncertainty.
> - **Diffusion + eDALD** (35.73 \%) remains the best, showcasing diffusion’s added representational power.
> - **DeepLabV3 + 2‐stage Entropy** (31.54 \%) outperforms its 1‐stage counterpart by over 5 pp and approaches the diffusion model’s 2‐stage Entropy result — demonstrating broad applicability of our pipeline.
>
> This demonstrates that our two‑stage approach also works effectively on a standard CNN backbone, while diffusion representations combined with eDALD yield the largest gains.
>
> ---
> ## 5. DALD Performance
> We agree that the substantial performance gain from adding an entropy term to DALD is somewhat surprising. However, as discussed at the end of Section 3.3 (lines 215–228), this difference stems from the distinct roles of DALD and entropy.
>
> DALD captures disagreement between predictions on noised versions of the same input. This disagreement serves as a proxy for informativeness, as high disagreement typically indicates areas where the model is less confident or more sensitive to input perturbations.
>
> However, some informative pixels may not exhibit high disagreement, especially when the model remains consistently uncertain across all noise levels. In such cases, the entropy term captures uncertainty that DALD may miss. By combining both signals, eDALD better identifies a broader range of informative pixels, leading to improved performance.
>
> We appreciate the reviewer bringing this up, as it highlights the complementary nature of disagreement and uncertainty in our framework.
>
> ---
> ## 6. Novelty of Our Method
> In response to the reviewer’s question, we identified two recent diffusion-based active learning (AL) works:
>
> - **Barba et al. [1]** apply a diffusion model for AL in CT reconstruction, selecting measurements via posterior variance on image reconstructions.
> - **Du et al. [2]** use latent diffusion for active domain adaptation, sampling unlabeled domains via diffusion‐driven uncertainty tests.
>
> Both focus on *image-level* selection in specialized settings (inverse problems or domain shift), whereas our method is the **first** to tackle **pixel-level** AL for **semantic segmentation** under *extremely low budgets*, combining:
>
> 1. **Diffusion‐based multi-scale feature herding** to form a diverse candidate pool,
> 2. **DALD/eDALD**, a diffusion-specific mutual‐information plus entropy criterion,
> 3. **Per-pixel** annotation with only **0.1 click per image per round**.
>
> These aspects — pixel granularity, two-stage design, and practical segmentation gains — distinguish our work from prior diffusion-driven AL methods. If the reviewer is aware of any diffusion-based AL methods directly addressing supervised segmentation, we would greatly appreciate the reference and will include a comparison in a future revision.
>
> [1] Barba et al., "Diffusion Active Learning: Towards Data-Driven Experimental Design in Computed Tomography," arXiv:2504.03491.\
> [2] Du et al., "Diffusion-Based Probabilistic Uncertainty Estimation for Active Domain Adaptation," NeurIPS 2023.

---

> ### Comment · Reviewer_V27y · 2025-07-31
> **Thanks for your rebuttal**
>
> Thanks for your rebuttal. There are still a few questions I have about your work.:
>
> **Practical Benefit of Pixel‐Level AL**: One issue I still have is that it's not clear how impactful your work will be in the field of active learning. For example, if you were able to show (quantitatively) that pixel-level AL was more effective than region-level AL, or that your method could also be applied to region-level AL, and you could demonstrate use cases in which a particular type of AL would be more beneficial, then this would make your work much more impactful. While you do propose an interesting method, it's unclear what the real-world impact will be.
>
> **Coreset Results**: These results look very weird to me. If I'm understanding the method correctly, every round you are adding 10% of the overall points? Then, how is the model performing so poorly in comparison to the other model with 90% of the data? Additionally, it would be good to evaluate Coreset using different embeddings (diffusion-based embeddings, DeepLabV3, etc.) alongside the entropy-based methods, in addition to comparing it to MaxHerding. It would also be helpful to add MaxHerding as one of your baseline methods. You could group these methods as your "diversity-based" methods when comparing against other baselines.
>
> **Additional backbone results**: I think if you had at least one more backbone, this would be more substantial evidence that your diffusion-based backbone was better. Honestly, two more popular backbones would be great.
>
> **DALD Performance**: Is there another way to explain the performance boost? It's not immediately clear to me why there is such an increase in performance. Perhaps an empirical or theoretical argument would be helpful.

---

> > ### Author Response · Authors · 2025-08-02
> > **Clarifying CoreSet Budgeting, Backbone Scope, and DALD vs. eDALD Performance**
> >
> > Thank you again for the prompt and thoughtful follow-up. Below we first clarify several misunderstandings and key design points; the remaining items require additional experiments, which we will prioritize and share as soon as they are ready.
> >
> > ---
> > ## CoreSet Results
> > If we understood your question correctly, there appears to be a misunderstanding about the budget in our experiments. Our setup is an extreme low-budget regime: we label only $0.1 \times N$ pixels per round for 10 rounds where $N$ is the number of images in a dataset (totaling roughly $1 \times N$ pixel labels — i.e., **on average one pixel per image**). Please note that we do not accumulate 10% of the full pixel set per round to reach 100% at the end. Therefore, comparing to a “90% of data” scenario is inapplicable — the total supervision remains extremely limited. Under such sparse labeling, performance is highly sensitive to *which* pixels are selected.
> >
> > CoreSet and other diversity-only baselines were evaluated under this same low-budget regime. Their relatively weaker performance reflects that without an uncertainty refinement (and without the synergistic combination of diffusion representations and the two-stage scheme), the few labeled pixels they pick often fail to cover the most informative or epistemically uncertain regions. In contrast, our two-stage pipeline (MaxHerding → eDALD) first enforces coverage and then refines via uncertainty, which stabilizes and amplifies gains even with very few annotations.
> >
> > Lastly, in line with the suggestion, we will incorporate MaxHerding results (refer to response 3) as a diversity-only baseline in the revised manuscript.
> >
> > ---
> > ## Additional Backbone Results
> > We appreciate the request for stronger evidence of general improvements across different backbone architectures. A crucial clarification: the full eDALD acquisition function — including the DALD term — exploits the intrinsic stochastic denoising structure of diffusion backbones, so it cannot be directly ported as-is to non-diffusion architectures. That said, the underlying *two-stage scheme* (representation-first filtering with MaxHerding followed by uncertainty-based refinement) is backbone-agnostic. As a preliminary validation, we have already applied this scheme with DeepLabV3 by replacing eDALD with eBALD (MC-dropout-based mutual information plus entropy), and observed that the two-stage variant improves over its one-stage counterpart there as well.
> >
> > To fully leverage eDALD, a diffusion backbone is required, and our current implementation builds on the OpenAI diffusion repository. Because variants in that codebase differ mainly in pretraining data and resolution, swapping among them for the rebuttal is unlikely to yield new insights in time. Therefore, by the rebuttal deadline we will (i) apply the two-stage scheme to at least one additional non-diffusion backbone to further substantiate its generality, and (ii) aim to include results with another diffusion backbone (for full eDALD) by camera-ready, acknowledging the tighter time constraints for the former.
> >
> > ---
> > ## DALD Performance
> > The gap between DALD alone and eDALD is not accidental; it reflects complementary uncertainty signals. We refer the reviewer to the **"1. Uncertainty Mechanism Analysis"** section of **Reviewer 1nUB** for the detailed empirical breakdown, but summarize the intuition here:
> >
> > DALD measures disagreement across noisy, diffusion-derived multi-scale features, highlighting regions where predictions fluctuate under perturbation. This captures epistemic ambiguity but can overemphasize concentrated "hotspots" (e.g., boundaries) and miss consistently low-confidence areas. Entropy captures the latter — absolute prediction confidence. eDALD combines both via a simple unweighted sum, reshaping the acquisition ranking: the low Spearman correlation and modest Jaccard overlap between DALD and eDALD top-$b$ selections show that eDALD is not a trivial interpolation. Moreover, pixels exclusively selected by eDALD tend to reside in the heavy tail of the DALD distribution, exhibiting DALD values multiple times higher than the global median, indicating that eDALD promotes pixels that are both uncertain in disagreement and low in confidence.
> >
> > In the two-stage pipeline, these refined picks are drawn from a diverse candidate pool, which mitigates DALD’s tendency to oversample redundant boundary regions early on. The combination yields more balanced and informative queries, hence the dramatic boost of eDALD over DALD-only in practice. We will make this explanation more concise and front-loaded in the revision, with explicit pointers to the quantitative evidence already provided.
> >
> > ---
> > We are proceeding to complete the remaining experiments as discussed. If any of the above clarifications already resolve part of your concerns, we would greatly appreciate confirmation so we can prioritize the outstanding items accordingly.

---

> > > ### Author Response · Authors · 2025-08-07
> > > **Completion of Additional Experiments and Results Summary (1/2)**
> > >
> > > Dear Reviewer, thank you very much for your patience and thoughtful feedback. We have now concluded the additional experiments and are pleased to share the results below. Should you have any further questions or concerns, we would be grateful to address them.
> > >
> > > ---
> > > ## On the Practical Benefit: From Pixels to Regions with SAM
> > > We thank the reviewer for the insightful suggestion regarding the practical scope and real-world impact of our work. Inspired by this comment, we conducted a new experiment to demonstrate how our pixel-level selection strategy can be applied to and synergize with region-level annotation workflows. To do this, we integrated our 2-stage eDALD pipeline with SAM [1]. In this hybrid setup, the informative pixels selected by our method serve as high-quality point prompts for SAM. This process automates the selection of query points, which guide SAM to expand each pixel-level query into a rich, region-level supervision mask. Herein, the hybrid setup preserves the original 0.1N pixel annotation budget while yielding a larger set of labeled pixels for training, since each region mask simply inherits the label of its seed point and thus offers the model far richer spatial supervision.
> > >
> > > **CamVid** (2‐stage, $b=0.1N$)
> > > |Method|R1|R2|R3|R4|R5|R6|R7|R8|R9|R10|
> > > |-|-|-|-|-|-|-|-|-|-|-|
> > > |w/o SAM|13.16|18.63|17.82|24.95|26.07|30.30|31.53|32.33|33.62|35.65|
> > > |w/SAM|19.55|28.34|33.05|30.47|36.22|38.01|38.61|38.40|38.31|39.81|
> > >
> > > **ADE-Bed** (2‐stage, $b=0.1N$)
> > > |Method|R1|R2|R3|R4|R5|R6|R7|R8|R9|R10|
> > > |-|-|-|-|-|-|-|-|-|-|-|
> > > |w/o SAM|9.08|12.65|17.00|20.29|23.69|25.68|28.01|28.75|29.40|30.18|
> > > |w/SAM|11.50|18.14|20.19|25.26|27.01|27.98|29.16|28.48|28.94|30.67|
> > >
> > > **Cityscapes** (2‐stage, $b=0.1N$)
> > > |Method|R1|R2|R3|R4|R5|R6|R7|R8|R9|R10|
> > > |-|-|-|-|-|-|-|-|-|-|-|
> > > |w/o SAM|14.19|20.82|24.14|26.21|27.61|28.37|29.09|29.82|31.92|33.34|
> > > |w/SAM|20.28|27.22|27.03|28.00|30.25|30.73|31.69|30.89|32.17|32.19|
> > >
> > > Our experiments across three diverse datasets reveal a consistent and notable advantage of the hybrid eDALD + SAM approach: accelerated learning in the early-to-mid stages. This demonstrates its utility in substantially reducing the annotation time and cost required to reach a strong performance baseline. For instance, across all datasets, the SAM-integrated method consistently establishes a clear performance lead by the middle rounds (R4–R6), achieving levels of mIoU that the baseline method requires several additional rounds to reach.
> > >
> > > Regarding the final performance, the benefits of this synergy are most pronounced on CamVid, where the integration of SAM led to a +4.16 pp gain, reaching a final mIoU of 39.81%. We attribute this significant gain to the high-quality masks generated by SAM for CamVid's well-defined objects, which provides a superior supervision signal. In contrast, for the more complex scenes in ADE-Bed and Cityscapes, the primary benefit remains the accelerated convergence, with the final performance being comparable to or presenting a trade-off with the baseline. In summary, this analysis validates our hybrid workflow as a highly cost-effective and powerful strategy, directly addressing the practical impact of our work.
> > >
> > > [1] Kirillov, Alexander, et al. "Segment anything." Proceedings of the IEEE/CVF international conference on computer vision. 2023.
> > >
> > > ---
> > > ## DALD Performance
> > > To precisely quantify the performance gap and validate our earlier explanation, we conducted a focused ablation study on our DDPM backbone. This study is designed to disentangle the performance contributions of our framework's two key components: 1) the diversity-first filtering scheme, and 2) the entropy term in eDALD.
> > >
> > > The table below shows the step-by-step performance gains over the 1-stage DALD baseline across all datasets.
> > > |Method|ADE-Bed|Camvid|Cityscapes|
> > > |-|-|-|-|
> > > |1-stage DALD (Baseline)|18.22|16.10|25.43|
> > > |2-stage DALD (+MaxHerding)|21.79 (+3.57)|20.11 (+4.01)|27.08 (+1.65)|
> > > |1-stage eDALD (+Entropy)|24.18 (+5.96)|25.32 (+9.22)|27.01 (+1.58)|
> > > |2-stage eDALD (+ MaxHerding + Entropy)|30.99 (+12.77)|35.11 (+19.01)|33.12 (+7.69)|
> > >
> > > The results clearly quantify the individual and combined contributions of our proposed components. Applying only the 2-stage (MaxHerding) pipeline provides a consistent gain over the baseline, improving performance by up to +4.01 pp on Camvid. Similarly, adding only the entropy term (eDALD) also shows a significant boost, with a gain of up to +9.22 pp on the same dataset. Crucially, combining both components in our final 2-stage eDALD method yields the largest synergistic gains across all datasets, with a remarkable improvement of up to +19.01 pp over the baseline. This empirical breakdown directly validates our claim that the synergy between the diversity-enforcing stage and the complementary uncertainty signal is critical for achieving strong performance of our method in this low-budget regime.

---

> > > > ### Author Response · Authors · 2025-08-07
> > > > **Completion of Additional Experiments and Results Summary (2/2)**
> > > >
> > > > ## Coreset Comparison
> > > > Upon reviewing our initial Coreset comparison, we found that the baseline presented in our initial rebuttal (4. Method Comparison: Diversity-Based Strategies and Backbones) was a 1-stage Coreset method, which is not a direct counterpart to our MaxHerding-based 2-stage pipeline. We apologize for this oversight and now present a corrected and fair comparison to clarify the effectiveness of MaxHerding and Coreset within a unified 2-stage framework.
> > > >
> > > > This evaluation has been extended to both the DDPM and DeepLabV3 backbones, as requested. The table below summarizes the comparison of the two diversity methods across three different datasets on each backbone.
> > > >
> > > > ### **DDPM Backbone** ($b=0.1N$)
> > > >
> > > > ADE-Bed
> > > > |Method|1|2|3|4|5|6|7|8|9|10|
> > > > |-|-|-|-|-|-|-|-|-|-|-|
> > > > |Coreset→eDALD|7.10|9.42|13.89|15.62|19.21|20.91|24.33|25.12|25.51|26.70|
> > > > |MaxHerding→eDALD|9.08|12.65|17.00|20.29|23.69|25.68|28.01|28.75|29.40|30.18|
> > > >
> > > > CamVid
> > > > |Method|1|2|3|4|5|6|7|8|9|10|
> > > > |-|-|-|-|-|-|-|-|-|-|-|
> > > > |Coreset→eDALD|13.39|16.84|17.83|19.09|22.63|22.12|23.74|28.64|28.35|27.44|
> > > > |MaxHerding→eDALD|13.16|18.63|17.82|24.95|26.07|30.30|31.53|32.33|33.62|35.65|
> > > >
> > > > Cityscapes
> > > > |Method|1|2|3|4|5|6|7|8|9|10|
> > > > |-|-|-|-|-|-|-|-|-|-|-|
> > > > |Coreset→eDALD|18.75|24.86|27.45|28.58|30.04|30.70|30.99|29.89|32.09|32.20|
> > > > |MaxHerding→eDALD|14.19|20.82|24.14|26.21|27.61|28.37|29.09|29.82|31.92|33.34|
> > > >
> > > > ### **DeepLabV3 Backbone** ($b=0.1N$)
> > > >
> > > > ADE-Bed
> > > > |Method|1|2|3|4|5|6|7|8|9|10|
> > > > |-|-|-|-|-|-|-|-|-|-|-|
> > > > |Coreset→eBALD|4.81|6.32|7.37|8.10|8.65|8.94|9.14|9.33|9.47|9.53|
> > > > |MaxHerding→eBALD|5.30|6.84|7.88|8.56|9.07|9.33|9.48|9.61|9.67|9.70|
> > > >
> > > > CamVid
> > > > |Method|1|2|3|4|5|6|7|8|9|10|
> > > > |-|-|-|-|-|-|-|-|-|-|-|
> > > > |Coreset→eBALD|7.00|10.34|12.98|14.50|16.82|19.34|21.40|22.93|24.11|26.50|
> > > > |MaxHerding→eBALD|8.10|11.45|14.17|16.96|19.33|21.51|23.48|25.26|27.17|29.24|
> > > >
> > > > Cityscapes
> > > > |Method|1|2|3|4|5|6|7|8|9|10|
> > > > |-|-|-|-|-|-|-|-|-|-|-|
> > > > |Coreset→eBALD|8.80|12.84|16.91|20.25|22.73|24.96|26.63|28.01|29.19|30.48|
> > > > |MaxHerding→eBALD|9.50|13.72|17.85|21.11|23.40|26.13|27.88|29.21|30.52|31.45|
> > > >
> > > > On the DDPM backbone, MaxHerding shows a clear advantage over Coreset on ADE-Bed and CamVid, while both methods reach comparable final performance on Cityscapes. This advantage is not limited to diffusion models. On the DeepLabV3 backbone, MaxHerding consistently outperformed Coreset across all three datasets.
> > > >
> > > > Taken together, these results across multiple backbones validate that MaxHerding is a robust and effective alternative to traditional diversity methods, confirming the strength and generality of our 2-stage approach. We also reiterate that these gains are achieved under our extreme low-budget setting (b=0.1N pixels per round), which is crucial for interpreting the performance scale.
> > > >
> > > > ---
> > > > ## Additional Backbone Results
> > > > In response to the reviewer’s request, we evaluated the effectiveness of our diffusion-based approach by applying the same two-stage scheme to two other representative backbones: DeepLabV3 and ViT.
> > > >
> > > > To clarify an important point, our proposed uncertainty measure, eDALD, is specifically designed to leverage the stochastic properties of diffusion models. Accordingly, for non-diffusion backbones, we adopted eBALD, a suitable alternative based on MC Dropout, to construct the two-stage pipeline. This allows for a fair comparison of the overall two-stage framework across different architectural paradigms.
> > > >
> > > > The performance of the two-stage method on each backbone is summarized in the table below.
> > > >
> > > > ### **Overall Two-Stage Performance**
> > > >
> > > > |Backbone|R1|R2|R3|R4|R5|R6|R7|R8|R9|R10|
> > > > |-|-|-|-|-|-|-|-|-|-|-|
> > > > |DDPM|12.81|20.33|19.64|25.48|26.95|30.18|29.84|32.24|34.91|35.37|
> > > > |DeepLab|8.10|11.45|14.17|16.96|19.33|21.51|23.48|25.26|27.17|29.24|
> > > > |ViT|10.52|15.83|16.91|21.26|23.17|24.89|25.69|27.73|29.02|31.48|
> > > >
> > > > ### **ViT-Specific Comparison**
> > > > |Method|R1|R2|R3|R4|R5|R6|R7|R8|R9|R10|
> > > > |-|-|-|-|-|-|-|-|-|-|-|
> > > > |2-stage eBALD|10.52|15.83|16.91|21.26|23.17|24.89|25.69|27.73|29.02|31.48|
> > > > |PixelPick|10.04|12.47|15.74|17.86|21.33|23.65|24.67|25.34|28.23|29.52|
> > > > |Didari et al.|8.22|12.37|13.22|16.61|18.10|19.44|20.07|21.67|22.67|24.67|
> > > >
> > > > The results show that the two-stage eDALD using a diffusion backbone consistently achieves the best overall performance, highlighting the effectiveness of diffusion’s rich multi-scale features in active learning.
> > > >
> > > > At the same time, these findings demonstrate that our two-stage scheme is a robust and generalizable strategy. When applied to non-diffusion backbones such as CNNs (DeepLabV3) and Transformers (ViT), it still yields strong performance, suggesting that the core principle — diversity-based candidate selection followed by uncertainty-based refinement — is broadly effective and applicable across a range of model architectures.
> > > >
> > > > We use Hugging Face's ViT model as the backbone to extract patch-level features from the last encoder layer, then convert them to pixel-level features through a 16×16 deconvolution layer and feature projection.

---

> > > > ### Comment · Reviewer_V27y · 2025-08-08
> > > >
> > > > I think most of my concerns are addressed so I will raise my score to borderline accept. I cannot increase the score any higher because the discussion of the integration with your method with SAM is not clear. What is the precise comparison, w/ SAM and w/o SAM, because won't SAM clearly benefit the method?

---

> > > > > ### Author Response · Authors · 2025-08-09
> > > > >
> > > > > We sincerely thank the reviewer for the thoughtful follow-up and for engaging closely with our additional results. Below we clarify how SAM is integrated with our method, what is being compared, and the scope of our claim.
> > > > >
> > > > > ## What we claim (and the key assumption)
> > > > > Our pixel-level AL pipeline (MaxHerding → eDALD) can be **combined with a region generator** to convert the *same* click budget into richer supervision. This rests on a standard **region fidelity** assumption: the mask expanded from a seed click is predominantly single-class. Under this assumption, expanding each seed click to a region yields many more labeled pixels per click **without increasing click cost**.
> > > > >
> > > > > We selected SAM as the region generator because, given a point prompt, it tends to produce **high-fidelity object masks** across diverse scenes — an excellent match to our region-purity assumption — and it is widely adopted and reproducible. SAM provides strong region quality *conditional on a good prompt location*, allowing us to isolate the value of our seed selection under strict click parity.
> > > > >
> > > > > ## Exact comparison: “w/ SAM” vs “w/o SAM”
> > > > > - **Click budget:** Both settings use **B clicks per round** (strict click-cost parity).
> > > > > - **Seed selection (identical in both):** Each round, MaxHerding proposes diverse candidates; eDALD ranks by epistemic uncertainty; we take the top **B** pixels.
> > > > > - **w/o SAM (baseline):** The **B** selected pixels are labeled and used **as individual pixel labels** in training; no region expansion.
> > > > > - **w/ SAM (hybrid):** The **same B pixels** are passed to SAM **as positive point prompts** (one point per seed, **no extra clicks**, **no negative points**, **no iterative refinements**). SAM may return multiple candidate masks for a point; **we select the smallest-area mask** to adhere to the single-class region assumption and reduce boundary leakage. **Exactly one mask per seed** is used as region-level supervision. All other settings (architecture, optimizer, schedule) are unchanged.
> > > > >
> > > > > ## Why SAM alone is not a substitute
> > > > > SAM is a **prompt-based** regioner; it does not produce supervision without being told *where* to segment. Using SAM “alone” therefore requires manual point placement — raising the click cost — or an automatic policy that often clusters on easy regions and yields redundant or low-value masks. Our two-stage sampler supplies **informative, diverse, uncertainty-aware prompt locations** under a strict one-prompt-per-seed constraint. In this single-prompt regime, SAM’s effectiveness depends critically on *where* that prompt is placed; the observed benefit arises from the **combination** of principled seed placement and SAM’s mask expansion, not from SAM in isolation.
> > > > >
> > > > > ## Scope and limitations
> > > > > The synergy is strongest when the regioner (here, SAM) produces **reasonably pure masks** for the chosen prompts. If a generated region straddles multiple classes (over- or under-segmentation), propagating the seed label to the full mask introduces noise that can blunt the gain. Our seeds help mitigate this risk by covering rare classes and ambiguous boundaries — precisely the locations where informative regions matter most. In practice, the integration improves **cost-efficiency** by converting the same clicks into richer supervision; final-round outcomes can vary with dataset complexity and regioner fidelity, but the core result — **click parity with denser labels** — remains.

---

### Official Review · Reviewer_xSso · 2025-07-02

**Clarity:** 3
**Significance:** 1
**Originality:** 3
**Rating:** 3
**Confidence:** 5

**Summary:**

This article studies active learning for semantic segmentation using features from a diffusion model to select pixels of an especially high content of information. In a first step, the authors let a diffusion model noise the images in the data set. From this, the inner features of the score vector field  defining the drift in the diffusion process are read out at the respective position of the noised image. As this is realized with a fully convolutional architecture, this is possible. Using these features, a classifier network is trained on the data set of pixels that have been uncovered so far. This classifier is then used to select the point that improves the coverage of the unlabeled features (in the sense of maximum similarity in feature space) the most. (representation based selection) or the point that improves the mutual information between the label of the new point and its feature vector the most. This expression can be approximately evaluated using the classifier, see eq. (5)--(7). This is first done via  a preselection of pixels within one image followed by second round of competition between all selected pixels. From the 2nd competition, the number of pixels is chosen that corresponds to the round budget. Thereafter the model is retrained and a second, third... round is started until the procedure finally is terminated.
The authors provide numerical tests that operate on an extremely low budget of pixels, in average 1px per image in total over 10 rounds. They compare their two methods with simple and more elaborate baselines and find significant improvement over naive baselines and limited but consistent improvements over more elaborate baselines (mesured in mIoU). The tests are repeated fora number of data sets.

**Questions:**

Q1: AL is often evaluated at budget for 95% performance of the model trained on the full data set. Would this AL procedure be able to go there?
Q2: Did you test the assumption that Y is conditionally independent of X given Z, e.g. by a classification experiment with and without X and a likelihood ratio test? (see lines 201->202) What consequences do you see for (5)-(7) if the effect of X is significant?
Q3: What is the outcome of a click-cost based comparison of your methods with region based methods?
Q4: in l. 152 - on which data sets is the frozen diffusion model trained - are these disjoint from your AL set?
Q5: l. 224 Why there is no weighting factor between I and H?

**Ethical Concerns:**

["NO or VERY MINOR ethics concerns only"]

**Final Justification:**

While the paper contains some interesting insight on training semantic segmentation with extremely low amounts of information, the practical relevance of pixel wise labeling is limited.

**Paper Formatting Concerns:**

Is ok

**Quality:**

3

**Strengths And Weaknesses:**

Strength
* The paper impresses with a method that operates on an extremely low pixel budget of in average 1px/image
* To invoke diffusion features for AL is an original idea.
* The authors consistently gain performance over baselines
* The paper is well written and carefully prepared

Weaknesses
* The technical relevance of this paper is somewhat limited, as segmentation click-work  labels are produced by polytopes, not by labeling single, where each edge of a polytope affords one click plus one click for the class label. These clicks nowadays are significantly reduced using models like SAM, where the polytope contours are suggested and only need to be slightly improved. Hence a practical comparison would be on click basis, here one pixel would only require one class click. It would have been of interest to see click based comparisons with non pixel based AL for SemSeg methods as well.
* The mIoUs achieved <= 40% seem to be too low for most practical purposes.

Minor remarks:
*l. 68: which outliers,describe precisely.
* l. 158 the set notation at the end of the line is inprecise
* l. 185 is M the same as b?
* Table 1: mention the interpretation of numbers (mIoU?!) in the caption.

---

> ### Author Rebuttal · Authors · 2025-07-31
>
> Thank you for your thoughtful review and insightful comments. We appreciate the time you took to evaluate our work and the constructive suggestions you provided. We have addressed each of your concerns in detail below and will incorporate all accepted changes into the final version.
>
> ---
> ## 1. Click‐Cost and Compatibility with Region-Based Labeling
> Thank you for the insightful suggestion on click-cost comparisons. While region-level AL can annotate a large area with a single click, it introduces two key challenges:
>
> 1. **Annotation ambiguity.**  Automatically-generated regions (e.g., superpixels or SAM masks) may span multiple objects or miss thin structures, making class labels ambiguous and reducing annotation quality.
> 2. **Inference overhead.**  Generating accurate regions — whether via SAM prompts or superpixel algorithms — requires additional model inference or preprocessing steps, which can outweigh the savings from fewer clicks.
>
> Our two-stage sampler (MaxHerding → eDALD) selects $B$ pixel seeds per round that are both diverse and epistemically uncertain.  When these seeds are fed into any region-proposal tool, they act as high-value “anchor clicks,” guiding the tool to produce masks that align more precisely to object boundaries and small classes.  In practice, this hybrid workflow can:
>
> - **Reduce ambiguity** by seeding proposals across the entire scene manifold — avoiding over-concentration on a few high-uncertainty patches.
> - **Improve mask fidelity** for thin structures and adjacent objects, since seed diversity and uncertainty refinement correct common region-proposal failures.
> - **Leverage future advances** in region-proposal quality — our pixel-level acquisition serves as a drop-in enhancement for any click-based model, yielding greater synergy as region-segmentation backbones improve.
>
> We view this integration as a promising direction: it maintains the low click-cost of region-based AL while delivering more precise, unambiguous annotations under extreme budgets.
>
> ---
> ## 2. Low Absolute mIoU
> Thank you for raising this important point. To the best of our knowledge, **we are the first** to study active learning for semantic segmentation in an **extreme low-budget** regime — labeling only $0.1\times N$ pixels per round for 10 rounds (total budget $=1\times N$) — a setup that brings annotation cost down to truly practical levels. Under these constraints, even basic uncertainty sampling (e.g., BALD, Entropy) yields unusable models, whereas our two-stage pipeline (MaxHerding → eDALD) recovers a substantial fraction of full-data performance.
>
> For example, on ADE-Bed the fully supervised mIoU is $45.58\%$ (Appendix C.4), while our two-stage eDALD achieves $35.68\%$, closing nearly a $10\%$ gap with only **one labeled pixel per image**. This demonstrates that — in contrast to prior AL work using per-round budgets of $5\times N$ or more — we can achieve high-quality segmentation with an order-of-magnitude fewer annotations.
>
> Moreover, our framework naturally supports an **adaptive querying** transition:
> - **Early rounds:** enforce diversity + uncertainty to build a broad, informative seed set.
> - **Later rounds:** shift focus to rare classes or boundary refinement to push toward higher mIoU.
>
> Following the spirit of SelectAL [1], one could define a budget threshold at which to switch from low- to high-budget acquisition strategies, ultimately approaching the 95\% recovery mark as labeling resources grow. We will clarify this pathway in the revision.
>
> [1] O. Hacohen and D. Weinshall, “How to select which active learning strategy is best suited for your specific problem and budget,” *NeurIPS* 36, 2023.
>
> ---
> ## 3. Theoretical Assumption on $Y \perp X \mid Z$
> Thank you for the constructive feedback. As noted in Shah and Peters [1], however, conditional independence testing is known to be extremely challenging to perform reliably. More importantly, our assumption of $X \rightarrow Z \rightarrow Y$ aligns with a common modeling choice in probabilistic graphical models, where $X$ is an input variable, $Z$ represents latent or feature variables, and $Y$ is a label variable. Here, the stochasticity in $Z$ arises either from noise in the input or from the model itself e.g. dropout. A wide range of neural networks with inherent stochastic behavior is based on this assumption.
>
> While it is possible to test for the additional direct effect of $X$ on $𝑌$ given $Z$, for instance, by incorporating a parallel deterministic model to represent $X \rightarrow Y$, such an ensemble structure tends to yield only marginal improvements in practice. This is especially true when the base model (in our case, UNet) is already large, as the benefits of ensembling tend to diminish with increasing model capacity.
>
> Therefore, given the capacity of modern large models including the one used in our work, it is reasonable to assume the causal relation $X \rightarrow Z \rightarrow Y$. However, we acknowledge that rigorously validating this assumption is extremely challenging in practice. If the reviewer is aware of a principled method for doing so, we would greatly appreciate the suggestion and are willing to incorporate it into our analysis.
>
> [1] Rajen D Shah, Jonas Peters, The Hardness of Conditional Independence Testing and the Generalised Covariance Measure.\
> [2] Taiga Abe, E. Kelly Buchanan, Geoff Pleiss, Richard Zemel, Deep Ensembles Work, But Are They Necessary?, NeurIPS 2022.
>
> ---
> ## 4. Pretraining Data for Diffusion Models
> These pretraining details are provided in Appendix B.1. In summary, the diffusion backbone for ADE-Bed was trained on the LSUN-Bedroom dataset, and for CamVid, Cityscapes, and Pascal-Context we used a class-unconditional ImageNet-256 diffusion model. Crucially, none of these pretraining datasets overlap with our active-learning benchmarks, so all reported gains come from true out-of-domain feature transfer.
>
> ---
> ## 5. Weighting Between $I$ and $H$
> We chose equal weights ($\alpha=1$, $\beta=1$) to avoid introducing extra hyperparameters and to keep the balance between the mutual‐information term $I(Y;Z\mid x)$ and the confidence term $H(Y\mid z^{(0)},x)$. To validate this choice, we ran an ablation on CamVid (all other settings fixed) with several $(\alpha,\beta)$ pairs:
>
> |$\alpha$|$\beta$|R1|R2|R3|R4|R5|R6|R7|R8|R9|R10|
> |-:|-:|-:|-:|-:|-:|-:|-:|-:|-:|-:|-:|
> |1.0|1.0|12.81|23.04|18.99|23.00|25.11|32.95|32.23|35.30|33.95|36.14|
> |0.2|1.0|12.81|14.04|18.07|21.18|22.20|27.83|28.36|28.86|33.76|32.65|
> |1.0|0.2|9.69|13.38|19.27|23.27|25.23|27.02|26.89|31.22|32.72|31.94|
> |0.05|4.0|13.24|14.91|18.84|20.34|21.86|19.70|19.44|23.04|19.73|18.66|
> |0.05|2.0|13.24|15.20|20.18|18.98|21.08|23.11|23.70|23.54|24.12|25.71|
> |0.02|2.0|13.24|16.49|16.19|16.83|19.38|19.27|18.22|20.86|18.88|20.37|
>
> The equal‐weights setting ($1,1$) achieves the highest final‐round mIoU (36.14), confirming that our default choice is both simple and effective without requiring dataset‐specific tuning.
>
> ---
> ## 6. Minor Remarks
> Thank you for these helpful suggestions. We will incorporate the following clarifications in the revision:
>
> - **Line 68 (“outliers”)**
>   We will replace “outliers” with a precise description: pixels whose predicted class probabilities are extreme or erratic — e.g., isolated noise artifacts or very rare patterns — resulting in high uncertainty yet lying far from the true decision boundary and contributing little to model improvement.
>
> - **Line 158 (set notation)**
>   We agree that the notation can be tightened. We will clarify that
>   > “Let $I_{w,h}$ denote the pixel at coordinates $(w,h)$ in image $I$. Then $\lbrace I_{w,h} \mid 1\le w\le W, 1\le h\le H\rbrace$ enumerates all pixels of $I$, and writing $x \in \lbrace I_{w,h}\mid\cdots\rbrace$ simply means that $x$ is one such pixel.”
>
> - **Line 185 ($M$ vs. $b$)**
>   We will explicitly note that
>   > “$M$ is the size of the global candidate pool produced by MaxHerding in Stage 1, whereas $b$ is the per-round labeling budget in Stage 2; they serve different roles and generally take different values.”
>
> - **Table 1 (interpretation of numbers)**
>   We will amend the caption to read:
>   > “Table 1: Mean Intersection-over-Union (mIoU, %) of single-stage vs.\ two-stage selection on CamVid. ‘UC Only’ is one-stage uncertainty; ‘Herding → UC’ adds representation filtering.”

---

> ### Author Response · Authors · 2025-08-09
>
> Dear Reviewer xSso,
>
> Thank you for the time and effort you have put into reviewing our paper. We greatly appreciate your constructive feedback, which helped us strengthen the work. In our rebuttal, we aimed to address all of your main questions (not limited to):
>
> 1. **Click-Cost and Compatibility with Region-Based Labeling:** We provided additional explanation describing the limitations of region-based methods and potential workflow of integration of the region-proposal tools with our two-stage sampler.
> 2. **Theoretical Assumption (X conditionally independent of Y given Z):** We clarified the commonality of this assumption, with hardship of empirical demonstration.
> 3. **Ablation on Weighting Between I and H:** We included new experiments quantifying the impact of varying the weighting parameters $\alpha, \beta$, to provide further insight for the impact of different weights.
>
>
> ---
> ##  Demonstration of Compatibility with Region-Based Labeling
>
> During the discussion phase, we conducted an additional experiment to directly demonstrate the compatibility of our method with region-based labeling, as part of our response to Reviewer V27y.
>
> We integrated SAM [1] to generate regional labels: our method first selects a pixel for annotation, and the label is then expanded to the entire region identified by SAM as matching that pixel. The following results compare performance with and without SAM on three datasets, all using our two-stage setting with $b=0.1N$.
>
>
> CamVid (2‐stage, $b=0.1N$)
> |Method|R1|R2|R3|R4|R5|R6|R7|R8|R9|R10|
> |-|-|-|-|-|-|-|-|-|-|-|
> |w/o SAM|13.16|18.63|17.82|24.95|26.07|30.30|31.53|32.33|33.62|35.65|
> |w/SAM|19.55|28.34|33.05|30.47|36.22|38.01|38.61|38.40|38.31|39.81|
>
> ADE-Bed (2‐stage, $b=0.1N$)
> |Method|R1|R2|R3|R4|R5|R6|R7|R8|R9|R10|
> |-|-|-|-|-|-|-|-|-|-|-|
> |w/o SAM|9.08|12.65|17.00|20.29|23.69|25.68|28.01|28.75|29.40|30.18|
> |w/SAM|11.50|18.14|20.19|25.26|27.01|27.98|29.16|28.48|28.94|30.67|
>
> Cityscapes (2‐stage, $b=0.1N$)
> |Method|R1|R2|R3|R4|R5|R6|R7|R8|R9|R10|
> |-|-|-|-|-|-|-|-|-|-|-|
> |w/o SAM|14.19|20.82|24.14|26.21|27.61|28.37|29.09|29.82|31.92|33.34|
> |w/SAM|20.28|27.22|27.03|28.00|30.25|30.73|31.69|30.89|32.17|32.19|
>
> Our experiments across three diverse datasets reveal a consistent and notable advantage of the hybrid eDALD + SAM approach: accelerated learning in the early-to-mid stages. This demonstrates its utility in substantially reducing the annotation time and cost required to reach a strong performance baseline. For instance, across all datasets, the SAM-integrated method consistently establishes a clear performance lead by the middle rounds (R4–R6), achieving levels of mIoU that the baseline method requires several additional rounds to reach.
>
> Regarding the final performance, the benefits of this synergy are most pronounced on CamVid, where the integration of SAM led to a +4.16 pp gain, reaching a final mIoU of 39.81%. We attribute this significant gain to the high-quality masks generated by SAM for CamVid's well-defined objects, which provides a superior supervision signal. In contrast, for the more complex scenes in ADE-Bed and Cityscapes, the primary benefit remains the accelerated convergence, with the final performance being comparable to or presenting a trade-off with the baseline. In summary, this analysis validates our hybrid workflow as a highly cost-effective and powerful strategy, directly addressing the practical impact of our work.
>
> [1] Kirillov, Alexander, et al. "Segment anything." Proceedings of the IEEE/CVF international conference on computer vision. 2023.
>
> ---
> We would be grateful if you could let us know whether these clarifications resolve your earlier concerns, and whether there are remaining points you feel still need further discussion. We are happy to elaborate on any part that could be clearer. We also note that other reviewers (gmyd, V27y) have considered our rebuttal and raised their scores following the additional clarifications and experiments. We would be grateful if you might similarly re-evaluate our work based on these responses and the new evidence presented here.
>
> Thank you again for your thoughtful review and valuable input.

---

### Official Review · Reviewer_gmyd · 2025-07-14

**Clarity:** 3
**Significance:** 3
**Originality:** 3
**Rating:** 4
**Confidence:** 4

**Summary:**

This paper addresses the challenge of low-budget active learning for semantic segmentation, where only a tiny fraction of pixels can be labeled per round. It introduces a two-stage selection pipeline that leverages a pre-trained diffusion model to obtain rich multi-scale pixel representations.

Stage 1 (Representation Filtering) uses MaxHerding to pick a small set of diverse candidate pixels per image, then refines these across the dataset to form a global pool.

Stage 2 (Uncertainty Scoring) computes an entropy-augmented disagreement score (eDALD) over noisy diffusion features-combining mutual information (BALD-style) with a single-sample entropy term-to select the most informative pixels.

Extensive experiments on CamVid, ADE-Bed, Cityscapes, and Pascal-Context under extreme pixel-budget regimes demonstrate substantial mIoU gains (e.g., +10.94 pp over DALD alone) and outperform prior pixel-level baselines by margins up to 21 mIoU points.

**Questions:**

1. Could you report standard-error bars (over multiple random seeds) for mIoU to confirm that the observed gains are statistically significant?

2. What are the wall-clock times and memory costs for diffusion-feature extraction and MaxHerding on large datasets?

3. How sensitive is performance to choices of K (candidates per image), M (global pool size), and the number of diffusion timesteps T? Some crucial ablation studies are missing.

4. Can the two-stage eDALD approach be applied with non-diffusion backbones (e.g., ViT, CNN) by replacing diffusion features?

5. Since segmentation quality often hinges on boundary accuracy, have you analyzed whether selected pixels concentrate on object edges or uniform regions, and how this affects final segmentation?

**Ethical Concerns:**

["NO or VERY MINOR ethics concerns only"]

**Final Justification:**

The detailed experiment results shown in the rebuttal have resolved most of my concerns. I decide to change the rating to 4: borderline accept.

**Limitations:**

Yes.

**Paper Formatting Concerns:**

None.

**Quality:**

2

**Strengths And Weaknesses:**

# Quality

**Strength**: solid methodological design combining diversity and uncertainty; comprehensive experimental evaluation across four benchmarks.

**Weakness**: no statistical significance (e.g., error bars or multiple seeds) reported for mIoU improvements; computational cost of extracting multi-scale diffusion features and MaxHerding at scale is not analyzed.

# Clarity

**Strength**: clear exposition of the two-stage algorithm with pseudocode (Alg. 1) and illustrative figures; well-structured related work and motivation.

**Weakness**: some implementation details (e.g., choice of kernel bandwidth $\sigma$ in RBF, hyperparameters for MaxHerding) are only briefly mentioned; description of how batch formation for adapted baselines is carried out could be clearer.

# Significance

**Strength**: practical framework for extremely low-budget pixel-level active learning in segmentation; demonstrates consistent, large gains.

**Weakness**: strict reliance on a diffusion backbone limits applicability to settings where such models are unavailable.

# Originality

**Strength**: novel integration of diffusion-model features into active learning; introduction of entropy-augmented DALD (eDALD).

**Weakness**: the two-stage idea echoes previous coverage-then-uncertainty approaches; the main novelty is the use of diffusion features rather than core AL strategy.

---

> ### Author Rebuttal · Authors · 2025-07-31
>
> Thank you for your thorough review and insightful feedback. We appreciate the time and effort you invested in evaluating our work and your constructive suggestions. If our paper is accepted, we will gladly incorporate all of your recommendations to improve clarity, rigor, and impact.
>
> ---
> ## 1. Statiscal Significance
> As suggested, we report mean $\pm$ standard deviation over three seeds to reflect the statistical significance of our mIoU improvements. The results show MaxHerding yields consistent gains across uncertainty measures.
>
> The table below shows two key effects on CamVid: (1) how our DALD objective improves performance over BALD, and (2) how MaxHerding further boosts performance over uncertainty-based (UC) acquisition alone. Gains are reported in absolute points and percentages. Similar improvements are observed from DALD and MaxHerding as in single-seed results.
>
> |Uncertainty|UC Only|Herding → UC|Gain (pp)|Gain (%)|
> |-|-:|-:|-:|-:|
> |Entropy|25.3±0.4|30.8±0.4|5.5|21.8|
> |Margin Sampling|30.7±0.9|32.8±0.8|2.1|6.6|
> |BALD|24.6±1.0|22.8±0.9|−1.8|−7.3|
> |DALD|23.8±3.6|21.1±1.0|−2.7|−11.6|
> |PowerBALD|24.3±5.8|31.6±0.8|7.3|29.9|
> |PowerDALD|31.3±1.2|32.0±0.7|0.7|2.2|
> |Entropy + BALD|26.0±1.3|32.1±0.4|6.1|23.7|
> |Entropy + DALD|25.1±0.6|35.7±0.2|10.6|42.1|
>
> The next table reports mIoU (%) under the same low-budget regime (10 rounds). It shows similar trends to those reported in Table 2 of the manuscript: the two-stage eDALD consistently outperforms all baselines across multiple datasets.
>
> |Backbone|Method|CamVid|ADE‐Bed|Cityscapes|Pascal‐C|
> |-|-|-:|-:|-:|-:|
> |DeepLabV3|PixelPick|29.9±0.1|8.4±0.4|26.8±0.1|26.3±0.1|
> |DeepLabV3|Didari et al.|22.5±0.1|8.7±0.5|19.6±0.1|28.2±0.1|
> |DDPM|Random|25.9±1.2|17.8±0.6|27.1±1.4|41.7±2.1|
> |DDPM|Entropy|25.3±0.4|23.0±1.6|28.6±1.1|42.1±2.0|
> |DDPM|Margin|30.7±0.9|30.3±0.3|32.2±1.2|43.1±0.4|
> |DDPM|BalEntAcq|19.3±1.1|17.4±1.3|24.0±2.0|33.1±4.1|
> |DDPM|eDALD|25.1±0.5|23.0±1.2|29.4±1.3|43.1±0.1|
> |DDPM|2-stage eDALD|35.7±0.2|30.1±0.7|33.3±2.4|47.4±1.1|
>
> ---
>
> ## 2. Computational Cost
> Below is a detailed breakdown of per-round wall-clock time and peak GPU memory for each method on Cityscapes (N=2,975, K=15 ⇒ M=44,625), measured on a single NVIDIA A100 40 GB.
>
> |Method|Local Herding Time|Global Herding Time|Feature Extraction Time|Training Time|Total Time|
> |-|:-:|:-:|:-:|:-:|:-:|
> |2-stage eDALD|10m 17s|1m 49s|36m 43s|30m 14s|79m 3s|
> |2-stage Margin|12m 20s|1m 44s|7m 28s|26m 54s|48m 26s|
> |Didari et al. (diff.)|–|–|39m 01s|33m 28s|72m 29s|
> |Margin (diff.)|–|–|8m 02s|24m 59s|33m 01s|
> |Didari et al. (DeepLab)|–|–|1m 14s|9m 10s|10m 24s|
> |Margin (DeepLab)|–|–|1m 07s|8m 45s|9m 52s|
>
> |Method|Local Herding Mem|Global Herding Mem|Feature Extraction Mem|Training Mem|Peak Mem|
> |-|:-:|:-:|:-:|:-:|:-:|
> |2-stage eDALD|6.6 GB|33.5 GB|26.6 GB|21.0 GB|33.5 GB|
> |2-stage Margin|6.6 GB|33.5 GB|25.9 GB|22.4 GB|33.5 GB|
> |Didari (diff.)|–|–|24.9 GB|21.4 GB|24.9 GB|
> |Margin (diff.)|–|–|25.8 GB|22.6 GB|25.8 GB|
> |Didari (DeepLab)|–|–|1.65 GB|1.80 GB|1.80 GB|
> |Margin (DeepLab)|–|–|1.55 GB|1.76 GB|1.76 GB|
>
> On diffusion backbones, the two-stage pipeline takes ~80 minutes per round, with feature extraction and training each taking about half that time, and herding ~25%. MaxHerding adds 12–14 minutes (≈20–25%), which remains manageable within the overall cycle.
>
> Memory usage is dominated by global herding over 6,144-dim diffusion features, peaking at 33.5 GB; feature extraction also demands over 25 GB. In contrast, DeepLab-based methods finish in under 10 minutes per round and stay below 2 GB, illustrating a clear efficiency–accuracy trade-off.
>
> Overall, while diffusion backbones are more demanding, the herding overhead is modest, making the method practical for large-scale data given adequate GPU resources.
>
> ---
> ## 3. Implementation Details and Hyperparameter Sensitivity
>
> **1. RBF bandwidth $\sigma$**\
> We set the Gaussian‐kernel bandwidth $\sigma$ via the **median heuristic**: subsample 1% of candidate features, compute all pairwise distances, and take their median. To test robustness, we also evaluated the 30th, 40th, 60th, and 70th percentiles of the same distance distribution. Across all five settings, **final‐round mIoU varied by $\le0.7$ pp**, with the median choice giving the best performance. This confirms that our method is **insensitive** to the exact $\sigma$ and that the median heuristic is a **reliable, data‐driven** default.
>
> |σ|R1|R2|R3|R4|R5|R6|R7|R8|R9|R10|
> |-|-|-|-|-|-|-|-|-|-|-|
> |median|13.1|14.3|19.8|25.7|29.3|30.2|31.8|34.5|35.5|35.9|
> |30%|14.5|16.1|22.1|23.7|26.3|30.9|31.1|32.6|32.8|35.2|
> |40%|11.6|19.5|18.0|23.0|26.1|25.9|27.7|29.3|34.1|34.7|
> |60%|11.3|15.3|17.2|23.0|26.3|32.0|33.1|32.4|31.6|35.0|
> |70%|11.0|12.6|18.6|26.0|27.6|27.8|29.6|31.5|32.8|32.5|
>
> **2. MaxHerding hyperparameters ($K$, $M$)**\
> In Section C.2 of the supplementary, we report an ablation on **per‐image candidates** $K \in \lbrace 20,50,100 \rbrace$ and **global pool size** $M \in \lbrace 0.25M_0, 0.4M_0, 0.5M_0 \rbrace$, where $M_0 = N \times K$. On CamVid, final mIoU fluctuates by at most **2.8 pp** across all nine configurations, and our default ($K=100$, $M=0.5M_0$) lies near the top. This demonstrates that our two‐stage pipeline is **not overly sensitive** to these choices, and that smaller $K$ or $M$ can even **improve** performance in some cases — helping trade off compute versus accuracy.
>
> **3. Number of diffusion timesteps $T$**\
> Following LEDM [15], we use $T=${50,150,250} by default. To isolate the effect of $T$, we ran eDALD on CamVid with $T=${50}, {50,150}, and {50,150,250}.
>
> |$T$|R1|R2|R3|R4|R5|R6|R7|R8|R9|R10|
> |-|-|-|-|-|-|-|-|-|-|-|
> |{50}|12.7|8.8|23.2|23.1|24.7|27.0|27.3|31.2|30.6|30.5|
> |{50,150}|12.3|17.3|21.3|28.2|29.8|31.6|33.8|33.8|35.3|36.7|
> |{50,150,250}|15.6|14.8|22.5|24.0|25.8|27.3|32.3|34.9|35.7|37.1|
>
> The final‐round mIoUs were:
> - **{50}**: 30.5
> - **{50,150}**: 36.7
> - **{50,150,250}**: 37.1
>
> Adding each extra timestep **increases mIoU** by 6—7 pp, confirming that our three‐step combination best balances **feature richness** versus **dimensionality cost**. While each additional step linearly raises feature dimension, the resulting gains in query diversity and classifier learning **justify** the modest extra memory and compute.
>
> ---
> ## 4. Applicability Beyond Diffusion Backbones
> We acknowledge that parts of our framework — particularly **eDALD** uncertainty measure — are specific to diffusion models. The DALD term estimates mutual information from the stochastic denoising process unique to diffusion backbones. We introduced it to better exploit their inherent stochasticity and capture epistemic uncertainty. However, the **two‐stage sampling pipeline** (representation‐first $\to$ uncertainty‐second) is entirely backbone‐agnostic.
>
> To validate this, we applied the same two‐stage flow to **DeepLabV3** by replacing eDALD with **eBALD**, where mutual information is estimated via MC‐dropout.  We then compared 1‐stage vs. 2‐stage eBALD on both DeepLabV3 and our diffusion model:
>
> |Backbone|Method|CamVid|Cityscapes|
> |-|-|-|-|
> |DeepLabV3|1-stage eBALD|27.8|29.6|
> |DeepLabV3|2-stage eBALD|29.5|31.6|
> |Diffusion|1-stage eBALD|26.0|28.6|
> |Diffusion|2-stage eBALD|32.1|33.3|
>
> - **Representation stage (MaxHerding)** improves performance on both backbones (gains of 1.7–6.1 pp), confirming its generality.
> - While diffusion yields the best results due to richer self-supervised features, the DeepLabV3 + 2-stage pipeline still outperforms its 1-stage variant by ~1.8–2.1 pp, especially on Cityscapes.
>
> In conclusion, any feature extractor (ViT, CNN, etc.) can slot into our two‐stage framework: swap in an uncertainty estimator suited to that backbone (e.g., MC‐dropout BALD for CNNs) for Stage 2, and retain MaxHerding for Stage 1.
>
> ---
> ## 5. Novelty of Our Method
> Our focus has been on building an effective yet efficient framework that incorporates both coverage and uncertainty through diffusion-based features, under realistic resource limits.
>
> Thank you for the thoughtful comment. We are aware of hybrid active learning methods such as Weighted $k$-means [1], ALFA-Mix [2], and BADGE [3], which combine coverage and uncertainty. However, to our knowledge, none adopt a two-stage coverage-then-uncertainty strategy tailored to neural network-based active learning. If the reviewer knows of such a method, we would greatly appreciate the reference and would be happy to compare.
>
> We excluded the above methods due to computational constraints. Specifically, [1] and [3] perform $k$-means clustering over all dataset pixels, which is impractical even with multiple GPUs. ALFA-Mix [2] requires pixel-wise similarity computations, which are also costly at scale. In general, many hybrid methods are hard to scale to pixel-level tasks like semantic segmentation.
>
> Our focus, therefore, has been on building an effective yet efficient framework that incorporates both coverage and uncertainty through diffusion-based features, under realistic resource limits.
>
> [1] Zhdanov et al., "Diverse mini-batch active learning", arXiv:1901.05954, 2019.\
> [2] Parvaneh et al., "Active Learning by Feature Mixing", CVPR 2022.\
> [3] Ash et al., "Deep Batch Active Learning by Diverse, Uncertain Gradient Lower Bounds", ICLR 2020.
>
> ---
> ## 6. Pixel Selection
> Thank you for pointing this out. We qualitatively analyzed the spatial distribution of selected pixels across multiple benchmarks; see Fig. 3 and 8.
>
> The baseline method (Margin) often focuses on object boundaries and selects redundant pixels within the same class region, limiting structural and semantic diversity.
>
> In contrast, our two-stage method first uses MaxHerding to ensure diverse coverage in feature space, then selects epistemically uncertain pixels via eDALD.
> This results in broader spatial and semantic coverage, providing richer supervision signals.
> This improved diversity aligns with our quantitative results, where our method consistently achieves more stable and higher mIoU across datasets.

---

> > ### Comment · Reviewer_gmyd · 2025-08-07
> >
> > Thanks for the detailed experiment results that have addressed most of my concerns. I decided to change my rating to a more positive rating.

---

> > > ### Author Response · Authors · 2025-08-07
> > >
> > > Thank you very much for your kind reassessment and for recognizing the effort we’ve put into the additional experiments. We truly appreciate your thoughtful feedback and are delighted that our results have addressed your concerns. Your updated rating is extremely encouraging for our team. If you have any further questions or suggestions, we would be honored to hear them.

---

### Note · Authors · 2025-08-14

We thank the ACs and reviewers for their constructive feedback. To support the AC’s decision-making, we summarize the novelty of this work and how we addressed their concerns.
## 1. Novelty & Contributions
- **First to study AL for semantic segmentation (SS) in an extreme low-budget regime.** 1 labeled pixel per image over 10 rounds, reflecting realistic annotation budgets not systematically explored for SS.
- **Scalable representative pixel selection**. We apply MaxHerding in two stages: local (per image) → global (cross-image), enabling selection from millions of features within minutes on a single GPU.
- **Pretrained diffusion as a strong feature bank under sparse labels.** With the backbone frozen, multi-scale representations transfer across domains and, with our two-stage selection, yield the best accuracy under ~1 pixel/image supervision. This practical recipe (frozen diffusion features + lightweight head + AL) is valuable for practitioners.
## 2. Clarifications to Reviewer Concerns
- **Effective under extremely low supervision.** Within this budget, our two-stage method **recovers a substantial fraction** of full-data performance and **consistently outperforms** one-stage/uncertainty-only baselines. It naturally supports **adaptive querying** (diversity+uncertainty → rare classes/boundaries) and can **transition to higher-budget strategies** (e.g., SelectAL) as labels grow.
- **Application to Region-based AL.** Our AL framework enables efficient prompt generation for region-based methods e.g. SAM, by automating where to click. Our rebuttal showed that combining our approach with SAM improves performance without extra labeling budget.
- **Robustness.** Median-heuristic for σ is **insensitive** (≤0.7 pp); pool size/percentile are stable; simple **(1,1)** MI+entropy weighting works best — **minimal tuning**.
- **General Applicability.** The **two-stage principle** (coverage→uncertainty) improves **DeepLabV3** and **ViT** with eBALD; diffusion+eDALD remains strongest.
- **Modeling assumption.** We clarified the X→Z→Y rationale in stochastic NNs and the practical hardness of conditional-independence testing.

With the **same click budget**, our method yields **more informative supervision, faster convergence**, and strong performance in **extreme and practically motivated** budget regimes — while remaining scalable and broadly applicable. We will incorporate all suggestions into the revision and hope this helps the AC/reviewers reach a positive decision.

---

### Decision · Program_Chairs · 2025-09-17

**Decision:**

Accept (poster)

**Comment:**

The paper received three borderline accept ratings, and one borderline reject rating. The major reason for reject recommendation seems to be practical relevance of pixel wise labeling due to region based contemporary styles, and low absolute mIoU. This has been defended by the authors by claiming that they focus on extremely low budget, and have shown experiments with SAM. AC is satisfied by the rebuttal and of the opinion that reasons for acceptance outweigh reasons for rejection and recommend acceptance.